# ANIMALGS: 4D ANIMAL RECONSTRUCTION FROM MONOCULAR VIDEO WITH 3D GAUSSIAN SPLATTING

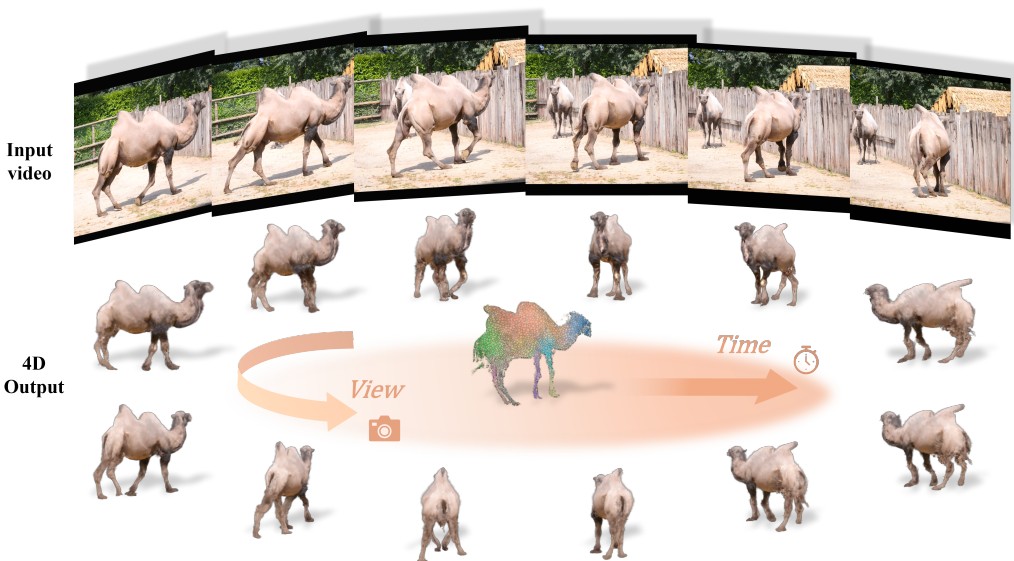

Figure 1: Given a monocular video of an animal (top), **AnimalGS** produces a high-fidelity, consistent 4D model (bottom), enabling free-viewpoint rendering across both time and viewing angles. Center: canonical 3D Gaussians colored by skinning weights.

## ABSTRACT

Reconstructing 4D animals from monocular videos is challenging due to large inter-species variation, complex articulations, and the lack of reliable templates. We introduce **AnimalGS**, a test-time optimization framework built on a 3D Gaussian Splatting representation for high-fidelity 4D reconstructions from single videos. Grounded in the insight that robust reconstruction emerges from pose-guided optimization rather than strict shape priors, AnimalGS treats priors as coarse initializations and integrates joint-aware and symmetry-aware designs to progressively disentangle motion and appearance. This leads to empirically strong generalization across diverse species and robustness to mismatching with shape priors. Extensive experiments demonstrate the superior performance of our approach [1] in geometry, motion, and temporal consistency across a wide variety of animal species.

## 1 INTRODUCTION

Animals in the natural world display a stunning diversity of shapes and behaviors. Accurately reconstructing their 3D shape and motion from visual data is crucial for various applications ranging from wildlife monitoring, animal conservation and ethology research, to immersive media content creation. Despite the wide accessibility of monocular video, the task of creating realistic 4D animal models from monocular video presents a significant challenge in computer vision. This is primarily

---

[1]Our code and results are to be published upon paper acceptance.

due to the inherent complexity of animal morphology and behaviors, as well as the fact that their appearance and motion are only partly observable from a monocular video.

The task of animal reconstruction presents unique challenges compared to 3D human reconstruction. Human models, such as SMPL (Loper et al., 2023; Pavlakos et al., 2019), benefit from well-studied anatomical structures and abundant 3D motion capture datasets. In contrast, animals of diverse species, ranging from camels, elephants to birds, exhibit extreme shape and motion variations, yet very little animal motion capture benchmarks are available. The pioneer SMAL model (Zuffi et al., 2017) is a parametric SMPL-like animal model learned from a limited collection of toy figurines; it captures a rather limited category of species and lacks realistic details of shape and motion. The dilemma of both scarcely labeled, partially observable animal data, and extraordinarily diverse shape & motion variations across animal species, forces a trade-off where, category-specific methods (Badger et al., 2020; Wang et al., 2021; Wu et al., 2023; Rueegg et al., 2022; Rüegg et al., 2023) achieve higher reconstruction quality by training on annotated dataset but struggle with generalization, while category-agnostic approaches (Li et al., 2024b; Aygun & Mac Aodha, 2024; Jakab et al., 2024) improve coverage at the cost of reconstruction fidelity. It motivates us to consider an alternative pathway of learning-based test-time optimization that does not require training from a labeled dataset, except for having a prior model to facilitate our initial shape reconstruction.

Extending to 4D reconstruction from monocular videos reveals a fundamental tension between representation flexibility and computational efficiency. Mesh-based methods (Yang et al., 2021a;b; Sabathier et al., 2024) are efficient but topologically constrained, while neural implicit methods (Yang et al., 2022; 2023a) offer flexibility at prohibitive computational costs. Recent 3D Gaussian Splatting approaches (Kerbl et al., 2023; Lei et al., 2024) provide a middle ground but still rely on parametric templates. Diffusion-driven methods (Ren et al., 2024a; Jiang et al., 2025) achieve impressive synthesis but sacrifice input fidelity. Existing methods either depend on rigid templates that limit generalization or generative priors that compromise reconstruction accuracy. We argue this trade-off stems from treating shape priors as strict constraints rather than flexible initializations.

Inspired by the above observations, we present **AnimalGS**, a pose-guided test-time optimization framework built on 3D Gaussian splatting for 4D single animal reconstruction from monocular videos. Our key insight is that robust 4D animal reconstruction is not dependent on highly accurate shape priors, which is contrary to common assumptions. Specifically, AnimalGS treats the animal shape prior as a coarse initialization and employs a hierarchical two-stage strategy: first, articulated motion is refined using joint-aware anchors together with a symmetry-aware temporal encoding that exploits bilateral cues to stabilize poses; second, non-rigid effects are captured via pose-guided deformation conditioned on global articulation context. This progressive disentanglement of motion, geometry, and appearance enables temporally coherent reconstructions across diverse species and behaviors.

In summary, our approach features the following key contributions:

- A novel test-time optimization framework is proposed, enabling 4D reconstruction of shapes and behaviors of a wide variety of animals from single monocular videos. This is achieved without access to well-annotated training dataset, or additional input requirement such as multi-view generative priors and category-specific shape templates.
- Our framework consists of two stages: a pose refinement stage followed by a pose-guided deformation stage (Figure 2). By introducing joint-aware anchors and symmetry-aware encoding, it progressively disentangles motion and appearance despite camera noise, enabling robust optimization even under inaccurate initialization.
- Extensive experiments demonstrate the superior quantitative and qualitative performance of our approach, across a wide variety of animal species, poses, and video types.

## 2 RELATED WORK

**3D Animal Reconstruction.** Reconstructing 3D animals is more challenging than reconstructing humans due to interspecific variation, complex articulations, and limited 3D data. Parametric models such as SMAL (Zuffi et al., 2017) provides the first skinned multi-animal template from toy figurines, following by various extensions and refinement (Zuffi et al., 2018; 2019; Rueegg et al., 2022; Rüegg et al., 2023). CSM-based methods (Kulkarni et al., 2019; 2020) predict dense image-to-surface mappings but remain tied to predefined templates. Template-free methods (Yao et al., 2022; 2023;

Liu et al., 2023a) discover parts and skeletons from sparse images through optimizing primitive part representation. Recent learning-based approaches scale to Internet data: UMR (Li et al., 2020), MagicPony (Wu et al., 2023), and Farm3D (Jakab et al., 2024) learn category-specific models, whereas FAUNA (Li et al., 2024b) and SAOR (Aygun & Mac Aodha, 2024) aim for category-agnostic reconstruction. The evolution in 3D animal priors has grounded a natural basis for 4D reconstruction, yet they are often treated as fixed constraints, and are severely limited in generalizing to unseen animal species. Instead, shape prior is engaged in our approach as merely coarse initialization, which has been empirically demonstrated to notably contribute to flexible and faithful 4D recovery across species.

**Dynamic Animal Reconstruction**    Extending static 3D models to capture temporal dynamics from monocular videos remains a central challenge in animal reconstruction. Deformation-based approaches (Yang et al., 2021a;b; Wu et al., 2022) represent objects by deforming an initial sphere mesh with fixed face connectivity, which struggle to recover fine surface details due to the limitations of template reliance.    Yang et al. (2022; 2023a) adopt NeRF-based representations for greater topological flexibility but suffer from prohibitive computational costs and lack explicit surface geometry. Hybrid explicit approaches (Sabathier et al., 2024; Lei et al., 2024) have recently emerged, among which GART (Lei et al., 2024) is highlighted for leveraging BITE initialization (Rüegg et al., 2023) and combining it with 3D Gaussian Splatting (Kerbl et al., 2023), allowing for more flexible shape representation. In spite of substantial progress, existing methods remain constrained by relying on fixed mesh topologies or category-specific templates, and typically fail to provide a systematic framework for jointly refining pose and shape under severe prior mismatches. In contrast, our method employs a progressive, pose-guided optimization strategy that allows the coarse prior to evolve during reconstruction, enabling robust adaptation to diverse animal species.

**Gaussian Splatting for Dynamic Scenes**    3D Gaussian Splatting (3DGS) (Kerbl et al., 2023) revolutionizes novel view synthesis by providing an explicit, flexible representation that achieves real-time performance with high fidelity. For dynamics, *time-augmented* 3DGS (e.g., 4DGS, Grid4D, Hybrid 3D–4D) encodes temporality in Gaussians but scales poorly with sequence length (Wu et al., 2024; Jiawei et al., 2024; Oh et al., 2025). *Deformation-based* variants keep a canonical 3DGS and learn warps (Deformable3DGS, SC-GS; spline extensions) for compact memory and smooth motion (Yang et al., 2024; Huang et al., 2024; Song et al., 2025). We adopt the deformation paradigm and drive warps with pose cues to improve temporal coherence for articulated animals.

**Video-to-4D Generation**    A parallel line of work leverages generative priors for video-to-4D synthesis. Zero-1-to-3 (Liu et al., 2023b) pioneered this direction by leveraging diffusion models to hallucinate novel views from a single image. Building upon Zero-1-to-3, DreamMesh4D (Li et al., 2024a) generates 4D models using a hybrid mesh-3DGS representation. In contrast, PAD3R (Liao et al., 2025) reconstructs 4D content based on Zero-1-to-3 by training a personalized PoseNet. SV4D (Xie et al., b) and SV4D 2.0 (Yao et al., 2025) extend this concept to videos, enforcing multi-frame and multi-view consistency. However, these methods struggle with long video sequences, often losing geometric detail and requiring fixed-length inputs. Other methods that directly generate 4D models, such as Splat4D (Yin et al., 2025), L4GM (Ren et al., 2024a), and GVF-Diffusion (Jiang et al., 2025) usually demand large model ensembles and high memory usage. They will also produce results with significant inconsistencies in both appearance and shape compared to the input video. In contrast, we pursue reconstruction-only supervision from the input video, avoiding generative mismatch while retaining faithfulness.

## 3    OUR APPROACH

Our goal is to recover time-varying 4D representations of animals from a monocular video sequence $\{I^t\}_{t=1}^{T}$ using a canonical-deformation formulation. A canonical 3D Gaussian Splatting model $G_{\text{can}}$ undergoes hierarchical transformation: first through articulated pose refinement ($G_{\text{pose}}^t$), then pose-guided non-rigid deformation ($G_{\text{deform}}^t$). The following sections detail initialization (Sec. 3.1), pose refinement (Sec. 3.2), deformation modeling (Sec. 3.3), and self-supervised optimization (Sec. 3.4), with the complete pipeline illustrated in Figure 2 with a cow video as an example.

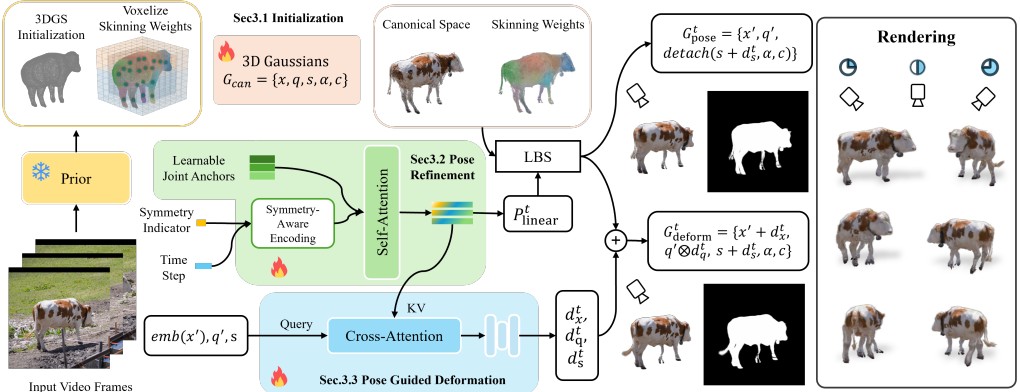

Figure 2: **AnimalGS** pipeline overview. From a monocular video input, an initial animal representation is constructed with the Fauna prior (Sec. 3.1), as 3DGS initialization, i.e. a canonical 3D Gaussian set $G_{\text{can}}$. This is then followed by two stages. The **Pose Refinement** stage integrates learnable joint anchors and time step encoded by a symmetry indicator to estimate articulated transformations, yielding intermediate representations $G_{\text{pose}}^t$ (Sec. 3.2). The **Pose-Guided Deformation** stage then predicts non-rigid displacements to obtain the final time-specific representation $G_{\text{deform}}^t$ (Sec. 3.3). **Right:** The resulting 4D representation can be rendered from arbitrary viewpoints and time steps.

## 3.1 INITIALIZATION FROM PRIOR

We initialize our canonical model using outputs from Fauna (Li et al., 2024b), which provides a coarse category-agnostic estimate of animal shape and pose from a single image. Given an input frame $I^t$, Fauna predicts $(V^t, W^t, C^t, P^t) = F(I^t)$, where $V^t \in \mathbb{R}^{N_v \times 3}$ are mesh vertices in rest pose, $W^t \in \mathbb{R}^{N_v \times J}$ are skinning weights, $C^t \in \mathbb{R}^{4 \times 4}$ is the camera pose, and $P^t \in \mathbb{R}^{J \times 4 \times 4}$ are joint transformations. Since vertices and skinning weights vary across frames, we use only the first-frame outputs $(V^1, W^1)$ for initialization and omit superscripts for clarity. Note that Fauna also predicts per-frame deformed meshes, but we discard them and retain only the coarse rest-pose shape for canonical initialization, as subsequent deformations are explicitly modeled within our framework. Inspired by Jiang et al. (2022); Lei et al. (2024), we further embed $W$ into a voxel grid to enable trilinear interpolation for dynamically created Gaussians during adaptive density control.

Following Kerbl et al. (2023), we represent canonical 3D Gaussian as $G_{\text{can}} = \{\mathbf{x}, \mathbf{q}, \mathbf{s}, \boldsymbol{\alpha}, \mathbf{c}\}$, where each Gaussian is parameterized by its center $\mathbf{x}$, orientation quaternion $\mathbf{q}$, scale $\mathbf{s}$, opacity $\alpha$, and view-dependent spherical harmonic coefficients $\mathbf{c}$. We initialize the Gaussian centers $\mathbf{x}$ from the rest-pose mesh vertices $V$, use a fixed initialization of 0.9 for opacity $\alpha$, and other initialization details can be found A.2.

## 3.2 POSE REFINEMENT

Initial per-frame poses $P^t$ predicted by the prior model are often unreliable due to limited views and articulation ambiguity in real-world videos. Our pose refinement module improves robustness by estimating per-joint transformations for linear blend skinning (LBS), mapping the canonical representation $G_{\text{can}}$ to posed states $G_{\text{pose}}^t$.

Instead of relying directly on noisy joint detections, we introduce learnable joint anchors that provide a stable articulation-aware representation. To incorporate bilateral symmetry, we design a **symmetry-aware temporal encoding** to handle camera estimation noise:

$$\mathbf{e}_t^{m,v} = \text{emb}(t \cdot m) \oplus v, \quad m, v \in \{-1, 1\}, \tag{1}$$

where $v$ indicates the visual flip state, $m$ identifies the camera reference group (accounting for the 3D drift between original and flipped estimated cameras), and $\oplus$ denotes vector concatenation. This encoding ensures flipped frames are treated as mirror-symmetric counterparts rather than independent temporal observations, enabling consistent use of symmetry cues (see Sec. 3.4 for details).

Anchors combined with $\mathbf{e}_t^{m,v}$ are processed by a self-attention block to produce joint-specific temporal features $\mathbf{F_J}^t \in \mathbb{R}^{J \times K}$, where $J$ denotes the number of joints and $K$ denotes the dimensionality of the keys and values in the following cross-attention block. These features are then projected to per-joint transformations $P_{\text{linear}}^t \in \mathbb{R}^{J \times 7}$ (quaternion rotation and translation) and also forwarded to the subsequent deformation stage, providing a temporal and joint-aware context. As illustrated in Figure 2, applying LBS with $P_{\text{linear}}^t$ updates Gaussian centers $\mathbf{x}$ and orientations $\mathbf{q}$, yielding the posed Gaussian $G_{\text{pose}}^t = \{\mathbf{x}', \mathbf{q}', \mathbf{s}, \boldsymbol{\alpha}, \mathbf{c}\}$.

### 3.3 Pose-Guided Deformation

While LBS-based pose refinement captures articulated motion, real animals exhibit complex non-rigid effects that cannot be modeled by skeletal transformations alone. Our pose-guided deformation module addresses this by predicting pose-conditioned, spatially varying offsets $(\mathbf{d}_x^t, \mathbf{d}_q^t, \mathbf{d}_s^t)$ to refine the Gaussian representation beyond articulation.

The key observation is that non-rigid deformations are tightly coupled with articulated pose, e.g. muscle bulging or skin motion varies with joint configuration. To capture this dependency, we adopt a cross-attention mechanism that conditions local Gaussian deformations on global joint-aware context. As shown in Figure 2, queries are constructed from Gaussian attributes—centers $\mathbf{x}'$ (with positional encoding), posed orientations $\mathbf{q}'$ and scales $\mathbf{s}$—while the joint-aware features $\mathbf{F}_J^t$ from the pose refinement module serve as keys and values. Using quaternion multiplication $\otimes$, the final deformed representation is $G_{\text{deform}}^t = \{\mathbf{x}' + \mathbf{d}_x^t, \ \mathbf{q}' \otimes \mathbf{d}_q^t, \ \mathbf{s} + \mathbf{d}_s^t, \ \boldsymbol{\alpha}, \mathbf{c}\}$. We deform only the geometric parameters while preserving appearance $(\boldsymbol{\alpha}, \mathbf{c})$, ensuring consistent texture and color across time.

### 3.4 Optimization

We optimize AnimalGS through *test-time optimization* on each input video. The overall objective is:

$$L_{\text{total}} = L_{\text{pose}} + L_{\text{deform}} + L_{\text{smooth}}, \tag{2}$$

where $L_{\text{pose}}$ corresponds to the pose refinement stage, $L_{\text{deform}}$ to the pose-guided deformation stage, and $L_{\text{smooth}}$ is a regularization term. We use a differentiable Gaussian rasterizer (Kerbl et al., 2023) to render RGB images $\hat{I}$, silhouettes $\hat{S}$, and normal maps $\hat{N}$ from $G$ with camera parameters $C$. Supervision combines photometric and silhouette objectives:

$$\mathcal{L}_{rgb}(\cdot) = (1 - \lambda_{\text{ssim}}) \, \mathcal{L}_1(\cdot) + \lambda_{\text{ssim}} \mathcal{L}_{ssim}(\cdot), \qquad \mathcal{L}_{sil}(\cdot) = \mathcal{L}_{bce}(\cdot) + \mathcal{L}_{dice}(\cdot). \tag{3}$$

with $\lambda_{\text{SSIM}} = 0.2$ in all experiments. All terms follow standard definitions.

**Pose Refinement Loss** The pose refinement stage uses silhouette-only supervision to isolate articulated motion from appearance, preventing texture artifacts from corrupting pose estimates while ensuring robust geometric alignment. Non-pose parameters are detached during rendering to block appearance-driven gradients, so that the posed Gaussian is represented as $\mathcal{G}_{\text{pose}}^t = \{\mathbf{x}', \mathbf{q}', \text{detach}(\mathbf{s} + \mathbf{d}_s^t, \boldsymbol{\alpha}, \mathbf{c})\}$, where adding the scale offset $\mathbf{d}_s^t$ can stabilizes downstream optimization without affecting pose gradients. The loss combines silhouette supervision with optional prior regularization:

$$L_{\text{pose}}^t = \lambda_{\text{pose}} \cdot \mathcal{L}_{\text{sil}}(\hat{S}_{\text{pose}}^t, S_{\text{SAM}}^t) + \lambda_{\text{prior}}(i) \cdot \|\hat{\mathbf{P}}_{\text{linear}}^t - \mathbf{P}^t\|_2, \tag{4}$$

where $\hat{S}_{\text{SAM}}^t$ are masks from Grounded-SAM (Ren et al., 2024b), $\lambda_{\text{pose}} = 0.2$, and $\lambda_{\text{prior}}(i) = \mathbf{1}_{[i \leq 4000]}$ provides early guidance, $i$ denotes the optimization iteration.

**Pose-Guided Deformation Loss** Unlike pose refinement, which focuses only on articulated alignment, the deformation stage jointly optimizes all Gaussian parameters—including appearance—to capture fine-grained details beyond skeletal motion. Given the deformed representation $G_{\text{deform}}^t$, we supervise both silhouettes and RGB renderings:

$$L_{\text{deform}}^t = \mathcal{L}_{rgb}(\hat{I}_{\text{deform}}^t, I^t) + \lambda_{\text{sil}} \cdot \mathcal{L}_{sil}(\hat{S}_{\text{deform}}^t, S_{\text{SAM}}^t), \tag{5}$$

where $\hat{S}_{\text{deform}}^t$ and $\hat{I}_{\text{deform}}^t$ denote the rendered silhouette and RGB image from $G_{\text{deform}}^t$, $\lambda_{\text{sil}} = 0.1$, and $I^t$ is the input frame. This stage ensures accurate geometry while recovering realistic textures and capturing non-rigid deformations.

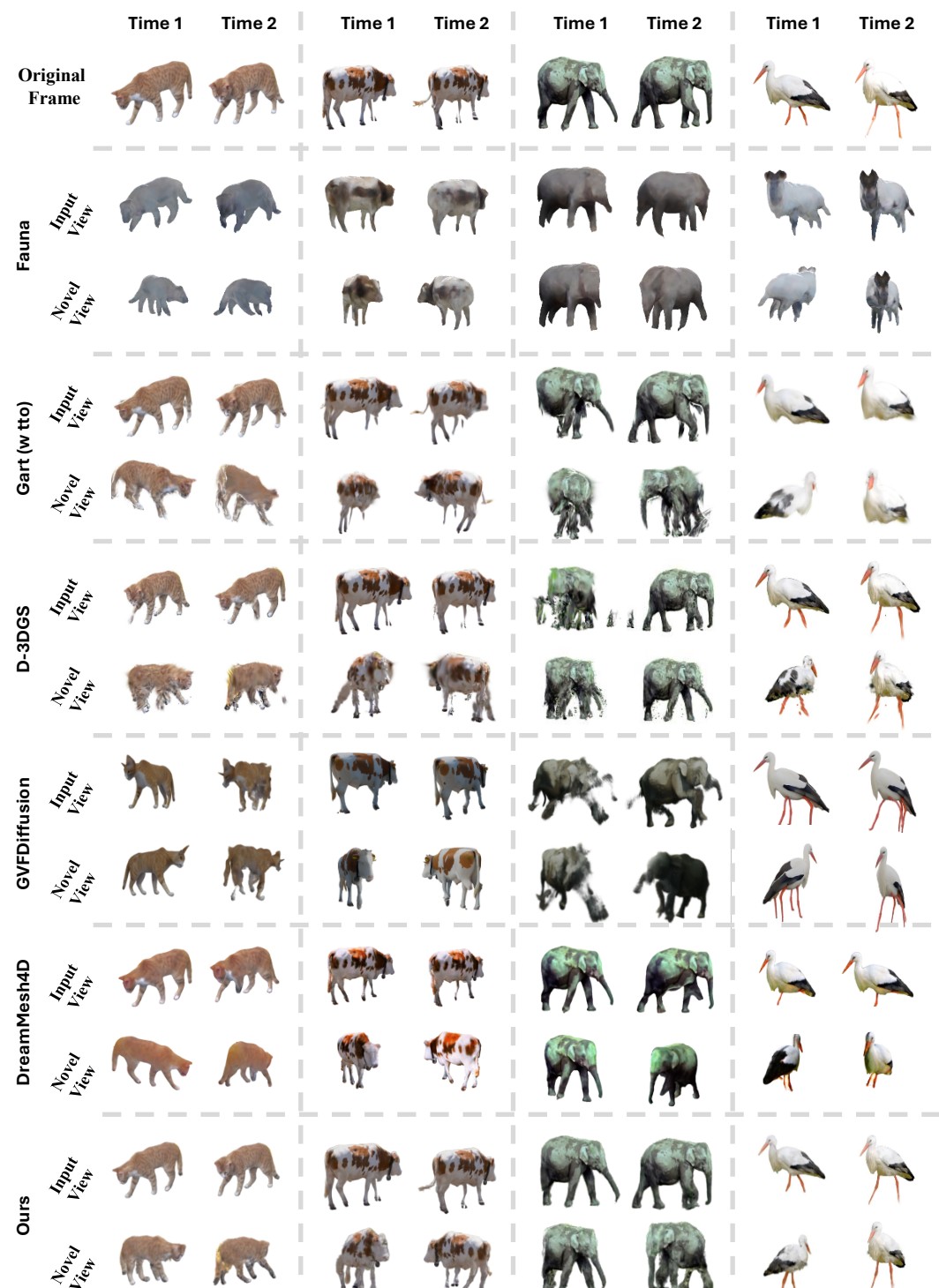

Figure 3: Visual comparison with the SOTA methods of Fauna (Li et al., 2024b), GART (Lei et al., 2024), D-3DGS (Yang et al., 2024), GVFDiffusion (Jiang et al., 2025), DreamMesh4D (Li et al., 2024a) at two randomly chosen time steps, t and t'.

**Smoothness Regularization**  To improve geometric stability, we regularize surface normals under random views of the deformed 3D Gaussians $G^t_{\text{deform}}$. For each rotated view $\theta$, we render a silhouette $\hat{S}^t_\theta$ and normal map $\hat{N}^t_\theta$, and penalize local angular variation using a total-variation style loss with

absolute cosine similarity to avoid inward and outward facing ambiguities.

$$\mathcal{L}_{\text{smooth}} = \lambda_{\text{smooth}}(i) \sum_{d \in \{x,y\}} \frac{\sum_{i,j} w_{i,j}^d \big(1 - |\hat{N}_\theta^t(i,j) \cdot \hat{N}_\theta^t(i + \delta_d, j + \delta_d')|\big)}{\sum_{i,j} \hat{S}_\theta^t(i,j) \cdot \hat{S}_\theta^t(i + \delta_d, j + \delta_d') + \epsilon}, \tag{6}$$

We use unit offsets $(\delta_x, \delta_x') = (0, 1)$ and $(\delta_y, \delta_y') = (1, 0)$, and $\lambda_{\text{smooth}}(i) = \mathbf{1}_{[7000 < i < 14000]}$ only activates at mid-training to avoid over-smoothing. This encourages view-invariant normal smoothness, reducing noise and flickering in novel-view renderings.

**Bilateral Symmetry Augmentation**   Naively treating horizontally flipped frames as independent samples discards their inherent geometric relationship. Meanwhile, imperfect camera calibration prevents strict symmetry enforcement, leading to ambiguity and conflicting supervision. To leverage symmetry without introducing noise, we build on the symmetry-aware encoding in Eq. 1 and construct four augmented samples for each frame $I^t$ under two geometric interpretations:

$$\mathcal{V}_{\text{orig}} = \{(I^t, C^t, P^t), (I_{\text{flip}}^t, C_{\text{sym}}^t, P_{\text{flip}}^t)\} \quad (m = 1), \tag{7}$$

$$\mathcal{V}_{\text{flip}} = \{(I_{\text{flip}}^t, C_{\text{flip}}^t, P_{\text{flip}}^t), (I^t, C_{\text{flip,sym}}^t, P^t)\} \quad (m = -1), \tag{8}$$

with symmetric cameras $C_{\text{sym}} = MC$ computed via sagittal plane reflection. Here, we set $v = 1$ for original frames $I^t$ and $v = -1$ for flipped frames $I_{\text{flip}}^t$.

This augmentation enriches supervision by exposing the model to both original and mirrored interpretations, while the symmetry indicators ($m$ and $v$) ensures consistent temporal encoding and suppresses calibration inconsistencies. As a result, bilateral symmetry is enforced effectively, leading to more stable geometry and motion reconstruction.

**Stabilization Strategies.**   We employ a pose blending with annealing scheme to enable stable refinement. The predicted pose $\hat{\mathbf{P}}_{\text{linear}}^t$ is blended with the prior pose to produce the final transformation:

$$\mathbf{P}_{\text{linear}}^t = w(i) \cdot \mathbf{P}^t + (1 - w(i)) \cdot \hat{\mathbf{P}}_{\text{linear}}^t, \tag{9}$$

where $w(i)$ anneals from 1 to 0 over 7K iterations. This allows gradual refinement from the initialization while enabling joint optimization of all modules from the start, preventing overfitting to inaccurate priors. In addition, we adopt the annealing smooth training mechanism (Yang et al., 2024), which injects decaying Gaussian noise into the temporal coordinate $t$ during early iterations. This improves temporal smoothness under pose inaccuracies without incurring extra overhead.

## 4   EXPERIMENT

### 4.1   DATASET

We collect 87 videos from three sources: online collection (11 videos), DAVIS (Perazzi et al., 2016) (8 videos), and APTv2 (Yang et al., 2023b) (68 videos). For APTv2, all videos contain 15 frames, except for two, which were manually composed by concatenating similar clips. We design a semi-automatic preprocessing pipeline: First, we extract animal masks using different strategies depending on the data source. For DAVIS, we directly use the provided instance masks and process each annotated instance independently. For the online collection, we apply Grounded-SAM (Ren et al., 2024b) with the animal category name as the text prompt and retain the bounding box with the highest confidence as input to SAM. For APTv2, we use the ground-truth tracking annotations as bounding-box prompts for SAM to ensure identity consistency across frames. Next, we compute smoothed bounding boxes and estimate animal and camera parameters for both original and horizontally flipped sequences using Fauna. Finally, we select temporally stable frames via DBSCAN clustering (Ester et al., 1996) on camera trajectories and sample every fifth frame for testing, resulting in a 4:1 train/test split.

### 4.2   IMPLEMENTATION DETAILS

We implement our method in PyTorch. Optimization is run for 20K iterations on DAVIS and online videos, and 10K on APTv2. Each iteration trains on $\mathcal{V}_{\text{orig}}$ or $\mathcal{V}_{\text{flip}}$ group. The symmetric augmentation only guide the final pose-guided deformation stage. All Gaussian parameters follow the learning rate

Table 1: **Input-view** quality on three datasets. Best in **bold**, second best underlined.

| Method | DAVIS | | | Online | | | APTv2 | | |
|---|---|---|---|---|---|---|---|---|---|
| | PSNR↑ | SSIM↑ | LPIPS↓ | PSNR↑ | SSIM↑ | LPIPS↓ | PSNR↑ | SSIM↑ | LPIPS↓ |
| Fauna (Li et al., 2024b) | 18.281 | 0.761 | 0.280 | 16.669 | 0.774 | 0.279 | 19.561 | 0.760 | 0.267 |
| D-3DGS (Yang et al., 2024) | 25.776 | 0.905 | 0.100 | 25.659 | 0.912 | 0.096 | 23.636 | 0.850 | 0.144 |
| GART (Lei et al., 2024) | 19.486 | 0.810 | 0.201 | 21.006 | 0.841 | 0.181 | 18.564 | 0.807 | 0.168 |
| GART (w/ tto) (Lei et al., 2024) | 21.347 | 0.859 | 0.171 | 23.128 | 0.883 | 0.158 | 19.167 | 0.834 | 0.150 |
| GVFDiffusion (Jiang et al., 2025) | 16.419 | 0.836 | 0.174 | 16.835 | 0.857 | 0.167 | 14.820 | 0.778 | 0.256 |
| DreamMesh4D (Li et al., 2024a) | 23.150 | 0.881 | 0.116 | 23.859 | 0.889 | 0.131 | **26.000** | **0.899** | **0.102** |
| Ours | **26.254** | **0.917** | **0.085** | **25.961** | **0.918** | **0.085** | 24.678 | 0.873 | 0.133 |

Table 2: **Novel-view** quality on three datasets. Best in **bold**, second best underlined. A dash indicates metric not reported by the method.

| Method | DAVIS | | | Online | | | APTv2 | | |
|---|---|---|---|---|---|---|---|---|---|
| | KID-16V↓ | FVD-F↓ | FVD-Diag↓ | KID-16V↓ | FVD-F↓ | FVD-Diag↓ | KID-16V↓ | FVD-F↓ | FVD-Diag↓ |
| Fauna (Li et al., 2024b) | 0.279 | — | — | 0.334 | — | — | 0.247 | — | — |
| D-3DGS (Yang et al., 2024) | 0.199 | 1192.015 | 980.747 | 0.231 | 1244.444 | 1268.110 | 0.262 | 1181.275 | 1112.902 |
| GART (Lei et al., 2024) | 0.216 | 1750.899 | 1675.388 | 0.233 | 1470.950 | 1547.561 | 0.230 | 1355.4969 | 992.195 |
| GART (w/ tto) (Lei et al., 2024) | 0.208 | 1680.705 | 1579.871 | 0.238 | 1473.927 | 1364.587 | 0.228 | 1134.661 | 948.274 |
| GVFDiffusion (Jiang et al., 2025) | 0.145 | 1872.192 | 1270.189 | 0.179 | 1673.419 | 1387.732 | 0.274 | 1715.471 | 1575.461 |
| DreamMesh4D (Li et al., 2024a) | 0.148 | 1154.038 | 2257.034 | 0.185 | 1518.163 | 2911.299 | 0.188 | **629.482** | **720.280** |
| Ours | **0.138** | **912.685** | **804.277** | **0.159** | **959.232** | **1148.435** | 0.170 | 991.472 | 930.146 |

Table 3: Ablation Study on our Online Collections and Artemis dataset. Best in **bold**, second best underlined.

| Method | Online | | | | | | Artemis | | |
|---|---|---|---|---|---|---|---|---|---|
| | Input View | | | Novel View | | | Multi View | | |
| | PSNR↑ | SSIM↑ | LPIPS↓ | KID-16V↓ | FVD-F↓ | FVD-Diag↓ | PSNR↑ | SSIM↑ | LPIPS↓ |
| random Init | 25.253 | 0.909 | 0.102 | 0.199 | 1071.278 | 1167.620 | 22.189 | 0.889 | 0.113 |
| w/o Deform | 24.522 | 0.898 | 0.095 | 0.173 | 1019.340 | 1355.544 | 23.037 | 0.931 | **0.071** |
| w/o Joint Anchors | 25.872 | 0.916 | 0.086 | 0.170 | 1025.439 | 1255.729 | 23.059 | 0.896 | 0.104 |
| w/o Symmetry Encoding | 25.666 | 0.913 | 0.094 | 0.158 | 965.699 | 1176.181 | 21.515 | 0.920 | 0.090 |
| w/o $L_{smooth}$ | 25.942 | **0.918** | **0.085** | 0.153 | 994.022 | 1203.356 | 22.215 | 0.859 | 0.130 |
| full | **25.961** | **0.918** | **0.085** | 0.159 | **959.232** | **1148.435** | **24.113** | **0.937** | **0.071** |

schedule of 3DGS, and other modules are optimized with a single Adam (Kingma & Ba, 2015) with an exponentially decaying learning rate from $8 \times 10^{-4}$ to $1.6 \times 10^{-4}$. Rendering speed scales with the number of optimized Gaussians; on the online collection, we achieve an average of 109.9 FPS, and on DAVIS, 110.7 FPS, measured on a single NVIDIA A6000 GPU at resolution $512 \times 512$.

## 4.3 RESULTS AND COMPARISONS

**Baselines** We compare to 5 state-of-the-art methods representing different paradigms: (1) **Fauna** (Li et al., 2024b): Learning-based single-image 3D reconstruction, also serving as our initialization prior; (2) **GART** (Lei et al., 2024): Test-time optimization using SMPL/SMAL priors for articulated 3D reconstruction from monocular video. We report both GART (optimized on training frames only) and GART (w/ tto), where the model is further refined on test frames, following their original evaluation protocol; (3) **D-3DGS** (Yang et al., 2024): Dynamic 3DGS with learnable deformation fields, using the same Fauna-based initialization and silhouette supervision; (4) **GVFDiffusion** (Jiang et al., 2025): video-to-4D generation using pretrained 3D diffusion models (Trellis (Xiang et al., 2025)). (5) **DreamMesh4D** (Li et al., 2024a): video-to-4D generation using pretrained image-to-3d model (Zero-1-to-3 Liu et al. (2023b) ) and optimize with hybrid mesh–3DGS representation. For fair comparison, we enhance GART and D-3DGS with flip augmentation, treating flipped frames as new timestamps to avoid shape aliasing (specifically, assigning timestamps $t \cdot v$ to flipped frames, consistent with the configuration in our "w/o symmetry encoding" ablation study).

**Quantitative Comparison** Table 1 reports the input-view reconstruction performance on DAVIS, Online, and APTv2, evaluated on test-set frames, while Table 2 presents the novel-view results.

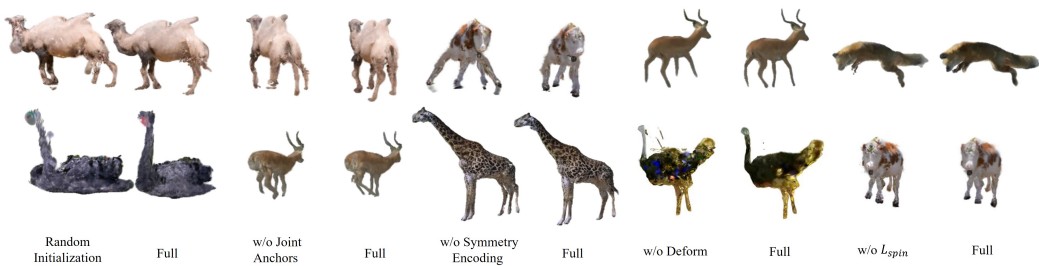

Figure 4: Novel-view synthesis under ablations, showing degraded reconstructions without certain components, while the full model remains stable and accurate.

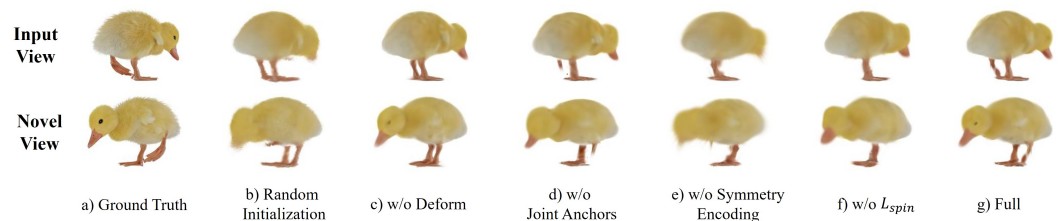

Figure 5: Ablation study on the Artemis dataset with available ground truth. The full model achieves the highest accuracy.

We adopt standard ground-truth metrics for evaluating input-view reconstruction. For novel-view quality, we follow Xie et al. (a) and use FVD-F (temporal coherence at a fixed view) and FVD-Diag (spatio-temporal consistency).

Since FVD-V requires equal numbers of frames and views—and therefore biases evaluation toward longer videos—we instead introduce KID-V (see Figure 9), which uniformly samples novel views and computes the Kernel Inception Distance (Bińkowski et al., 2018). KID-V is unbiased and remains stable even with limited samples, making it particularly suitable for our setting with only a few novel views per timestep. KID-V is computed on novel views from the test set, while FVD-F and FVD-Diag are evaluated over the entire generated sequence. For Fauna, we only report KID-16V because it is a single-image-based method.

Our method consistently outperforms all baselines on both input-view and novel-view metrics across DAVIS and Online. D-3DGS ranks second on input views but performs poorly on novel views, indicating overfitting to observed frames and difficulty maintaining consistent 3D geometry in unseen views. GVFDiffusion achieves the second-best KID scores on the DAVIS and Online collections, but struggles significantly on APTv2 due to its sensitivity to video length. DreamMesh4D performs well on APTv2 due to its short sequences (15 frames), large motion, and limited camera movement, where strong generative priors help compensate for sparse viewpoints. Nevertheless, our method achieves the second-best performance on APTv2 across all metrics. We further report the optimization time in Table 4. Our method achieves a balance between computational cost and reconstruction fidelity.

**Qualitative Comparison** Figure 3 shows reconstruction results across four species. Fauna (Li et al., 2024b) produces only coarse shapes with approximate color. GART (Lei et al., 2024) and D-3DGS (Yang et al., 2024) recover basic body structure and realistic appearance, but both fail to model limb geometry—GART completely misses the stork's legs, while D-3DGS separates limbs into disjoint parts. GVFDiffusion (Jiang et al., 2025) struggles with short sequences, producing distorted artifacts on APTv2 clips and hallucinating extra legs on the stork despite reasonable cow reconstruction. DreamMesh4D (Li et al., 2024a) maintains watertight surfacesbut exhibits structural inconsistencies in certain frames (e.g., the cat appears with only one leg at time 1), reduced appearance fidelity—especially under novel views—and unnatural motion, such as the stork's legs remaining rigidly stuck together. In contrast, our method achieves faithful, temporally consistent results across all species, preserving fine details in both input and novel views.

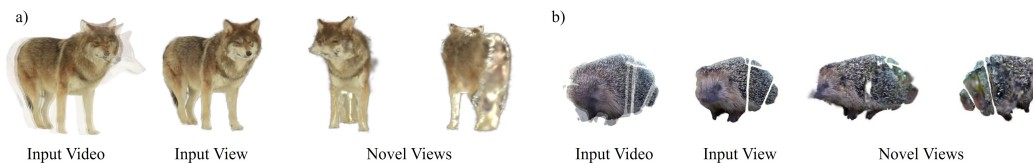

a)  Input Video    Input View    Novel Views    b)  Input Video    Input View    Novel Views

Figure 6: Limitations. a) Limited view coverage and unstable camera estimation cause localized artifacts such as drifting head geometry and vague rear legs. b) Persistent occlusion in the input video prevents faithful reconstruction of the hedgehog's body, leading to fragmented geometry.

**Ablation Study**    Table 3 quantifies the contribution of each component. Since no ground-truth novel views exist for the real videos in our Online Collection, the available metrics are only approximate: FVD mainly reflects temporal smoothness, while KID–16V and FVD cannot fully capture shape integrity or motion naturalness. This explains why numerical differences remain small despite the clear visual gap in Figure 4, where the full model produces noticeably better geometry.

To obtain more reliable measurements, we further evaluate ablations on Artemis (Luo et al., 2022), which contains 9 CGI animal species and 24 motions in total, captured by 36/72 synchronized cameras. We utilize dataset-provided cameras to evaluate the model components in isolation. Specifically, we optimize on single sampled monocular sequences and test on the remaining held-out views (training:test = 1:71 for wolf and 1:35 for the others). A comparison with our reimplemented D-3DGS is reported in Table 5. Figure 5 shows that our full model achieves the highest reconstruction accuracy.

While performance differences are minor on Online metrics, they become distinct on Artemis. Removing deformation causes the sharpest drop on Online, indicative of the nonlinear dynamics in real animals, while affecting rigid CGI assets less. Removing symmetry encoding yields the worst performance on Artemis, validating its necessity for structural consistency. Furthermore, several variants (Random Init, w/o Joint Anchors, w/o $L_{smooth}$) collapsed on 1-2 challenging sequences. These results highlight that our components jointly enhance accuracy, stability, and robustness.

### 4.4    DISCUSSION AND LIMITATION.

Our results highlight that treating animal priors as coarse initialization, rather than strict constraints, enables robust 4D reconstruction across species. Unlike recent generation-based approaches that prioritize plausibility over fidelity and degrade under dynamic deformation, our optimization-based framework emphasizes realistic reconstruction that remains closely aligned with the input video, even with severe prior mismatches. This highlights a critical insight for non-human reconstruction: balancing prior knowledge with optimization flexibility is key to realism and generalization.

As shown in Figure 6, our approach still struggles under limited viewpoints, inaccurate camera priors, or strong occlusions. Head motion is particularly sensitive to camera estimation noise. Unlike limb movements are more clearly constrained by the silhouettes, the head often exhibits subtle silhouette changes during rotation. This makes silhouette-based supervision insufficient to correct drift and causing localized artifacts. Future work could incorporate stronger geometric supervision (e.g., depth, keypoints) or learned deformation priors to address these limitations.

### 5    CONCLUSION

We presented AnimalGS, a test-time optimization framework for 4D animal reconstruction from monocular video. Our key insight that robust reconstruction arises from pose-guided optimization rather than accurate shape priors enables generalization without multi-view supervision or category-specific templates. By introducing joint-aware anchors and symmetry-aware encoding, our method disentangles motion from appearance in the presence of camera noise, and remains robust to prior mismatches. Extensive experiments across diverse species demonstrate clear improvements over state-of-the-art baselines in both reconstruction quality and temporal consistency, validated through quantitative metrics and user studies. While developed for animals, these principles may extend to other non-rigid objects, suggesting hybrid approaches that couple optimization precision with stronger geometric cues and multi-view synthesis.

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

# A APPENDIX

## A.1 THE USE OF LARGE LANGUAGE MODELS

In the process of preparing this paper submission, Large Language Models (LLMs) were used only as a writing-assist tool. Specifically, they were used to polish the text for better clarity and fluency, as well as to correct minor grammatical errors. The LLMs didn't contribute to the research ideation, methodology, or substantive writing of the paper. The authors take full responsibility for all the submitted contents.

## A.2 INITIALIZATION OF 3D GAUSSIAN ATTRIBUTES

For centers, we directly use mesh vertices. For orientations, we construct an orthonormal frame using the vertex normal and a randomly sampled tangent direction (standard normal distribution). For SH coefficients, the DC term is initialized with random RGB values (uniform distribution), and all higher-order SH terms are set to zero. We initialize scale based on vertex-connected edge lengths and use a fixed initialization of 0.9 for opacity.

Table 4: Comparison of optimization time (minutes) across different methods.

| Method | D-3DGS | GVFDiffusion | GART | GART (w/ TTO) | DreamMesh4D | **Ours** |
|---|---|---|---|---|---|---|
| Time (min) | 3.34 | 5.26 | 8.76 | 21.86 | 50.38 | 16.84 |

Table 5: Quantitative comparison results on Artemis with D-3DGS.

| Method | PSNR ↑ | SSIM ↑ | LPIPS ↓ |
|---|---|---|---|
| D-3DGS | 20.050 | 0.903 | 0.103 |
| **Ours** | **24.113** | **0.937** | **0.071** |

## A.3 EFFECT OF INITIALIZATION STRATEGIES

Figure 7 compares different initialization strategies for our 3D Gaussian representation. With prior-based initialization, the canonical space already provides a coarse but structured shape, leading to stable canonical 3DGS and skinning weights. In contrast, random initialization starts from an unstructured point cloud that produces unstable intermediate results in the canonical stage. Nevertheless, optimization can still converge to a configuration compatible with the skinning weights, although the process is less stable and less reliable than with prior guidance.

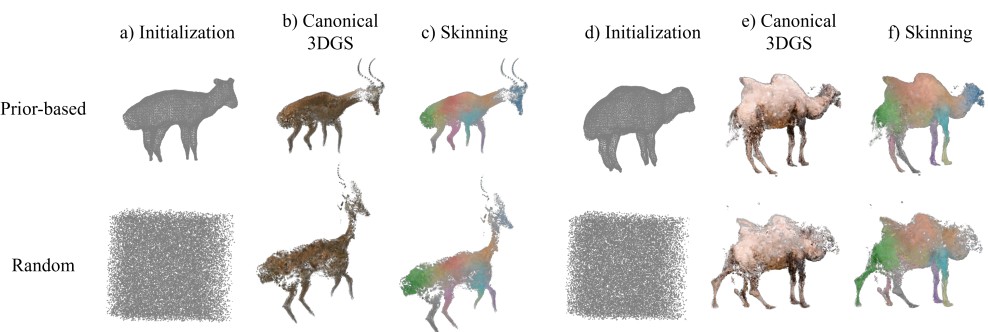

Figure 7: Effect of different initialization strategies.

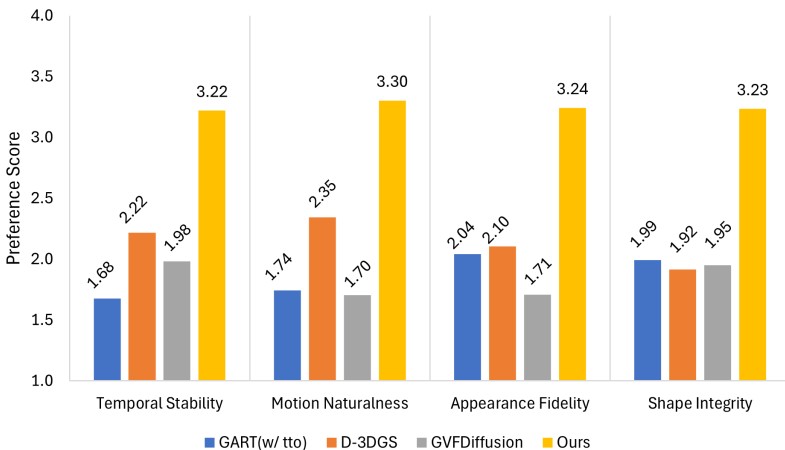

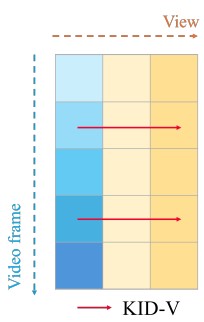

Figure 9: Illustration of KID-V metric.

Figure 8: User study preference scores across four perceptual dimensions.

## A.4 USER STUDY

Since existing quantitative metrics may not fully capture the perceptual quality of novel-view synthesis, we conducted a user study with 56 participants, evaluating reconstruction results of animal videos sampled from our three datasets. Among them, 8 participants had prior experience in 3D modeling, while the remaining participants had no such experience. Each participant was asked to rank four anonymized and randomly shuffled reconstruction methods (Ours, GART, D-3DGS, GVFDiffusion) across four perceptual dimensions: temporal stability, motion naturalness, appearance fidelity, and shape integrity. Rankings were converted to scores (4=best, 1=worst) and averaged. As shown in Figure 8, our method consistently achieves the highest preference scores across all dimensions, indicating superior perceptual quality compared to all baselines.

A total of 56 participants evaluated the reconstructed animal videos using a comprehensive rating system. Four methods were ranked from best to worst across four evaluation dimensions: Q1: 3D Temporal Stability (consistency of reconstructed shape and texture over time), Q2: Animal Motion Naturalness (realism of reconstructed movements), Q3: Appearance Fidelity (visual realism including identity consistency and appearance quality), and Q4: Shape Integrity (correctness and completeness of 3D geometry including structural soundness, completeness, and proportional accuracy). Each participant assessed four different methods based on one 4D GIF visualization for questions Q1 and Q2, evaluating the complete reconstructed animal, and 3D visualizations rendered at three distinct temporal moments for questions Q3 and Q4. All evaluation criteria (Q1-Q4) were presented with detailed explanations and examples to ensure consistent and informed participant assessments.

45 participants completed all evaluation questions; the remaining participants provided partial responses. To ensure fair analysis, weighted scores were calculated for each question based on the actual number of votes received. The evaluation process converted participant rankings into numerical scores (ranging from 4 for best performance to 1 for worst), which were then averaged to determine final ratings.

Table 6: Average and median Gaussian counts across datasets.

|  | DAVIS | Online | APTv2 | Artemis |
|---|---|---|---|---|
| **Average** | 45606.6 | 53323.8 | 43297.3 | 25649.9 |
| **Median** | 46155.0 | 37343.0 | 37003.0 | 24399.5 |

## A.5 GAUSSIAN COUNT ACROSS DATASETS

We report the average and median numbers of Gaussians used by our method across all datasets in Table 6. The notably lower Gaussian count on the **Artemis** dataset can be attributed to two main factors:

- **Accurate cameras.** Artemis provides ground-truth camera poses. Consequently, the optimization does not need to introduce additional Gaussians to compensate for reprojection errors caused by noisy camera estimation in real-video datasets.
- **Simpler motion and deformation.** Real animals exhibit complex non-linear soft-tissue dynamics and high-frequency local motion, which require a denser 3DGS representation. In contrast, the CGI assets in Artemis contain simpler non-rigid behavior, allowing accurate reconstruction using fewer Gaussians. This observation is consistent with our "w/o Deform" ablation, where removing the deformation module results in a much smaller performance drop on Artemis compared to real-video datasets.

