# OpenReview forum: "AnimalGS: 4D Animal Reconstruction from Monocular video with 3D Gaussian Splatting"
_ICLR.cc/2026/Conference — Submitted to ICLR 2026_

### Official Review · Reviewer_GMuu · 2025-10-26

**Soundness:** 3
**Presentation:** 3
**Contribution:** 2
**Rating:** 4
**Confidence:** 4

**Summary:**

This paper presents AnimalGS, a test-time optimization method for dynamic 3d animal reconstruction from a video, using 3DGS as the geometry representation.

It starts with a coarse shape from the 1st image by Fauna (Li et at 2024) and performs two-stage optimization: a pose refinement step using learnable joint anchors and symmetry augmentation, followed by pose-guided non-rigid deformation represented as Gaussian offsets.

It produces good 4D reconstructions across a variety of animal species, outperforming state-of-the-art methods such as GART, D-3DGS, and GVF-Diffusion.

**Strengths:**

- The method is technically solid and works better than the referenced existing works.
  - The use of symmetry prior by augmentation is interesting, although related ideas has been used in prior methods.
- Thorough evaluation that includes quantitative results and user studies.
  - The authors assembled a dataset of 87 monocular videos and reported both input and novel view metrics.
- The dynamic 3DGS representation allows fast rendering and interactive visualization.

**Weaknesses:**

- The paper has limited conceptual difference compared to existing literature. Test-time optimization, LBS-based deformation, and symmetry prior, and skinned 3DGS, are largely extensions of existing works.
- The improvement from introduced components (e.g., Tab 3) are not that significant.
- The visual results are not great. The reconstruction suffer from blurriness and appearance artifacts.
- Missing baseline and related work: it seems DreamMesh4d has more appealing result and should be compared against.

[A] DreamMesh4D. https://arxiv.org/abs/2410.06756

**Questions:**

- It would help if the authors could highlight the major difference compared to existing works in writing, such as LASR and DreamMesh4d.
- Compared to symmetry prior, image-to-3d models provides a stronger 3d prior and could help disentangle shape and motion.
- For evaluation, I found the current NVS metrics not convincing as those are distribution-level. Multiview dataset such as "Artemis: Articulated Neural Pets with Appearance and Motion Synthesis" can be an alternative to measure novel view view synthesis scores.

---

> ### Author Response · Authors · 2025-11-25
> **Response to Reviewer GMuu**
>
> Thank you very much for taking the time to provide such comprehensive and constructive comments on our study. Here are our answers to your questions:
>
> ---
>
>
> **W1**: *"The paper has limited conceptual difference compared to existing literature. Test-time optimization, LBS-based deformation, and symmetry prior, and skinned 3DGS, are largely extensions of existing works."*
>
>
> **A**: We thank the reviewer for raising this concern.While our method indeed builds on test-time optimization, LBS-based deformation, symmetry priors, and 3DGS, it is not a simple combination of them.
> Our main novelty lies in a test-time optimization framework specifically for high-fidelity 4D animal reconstruction from single video without labeled training data and multi-view generative priors; a dedicated model with a two-stage design, joint-aware anchors, and symmetry-aware encoding to progressively disentangle motion and appearance under noisy initialization; and the use of 3DGS representation avoids mesh topology constraints, converges faster in practice, captures high-frequency details (e.g., fur), and produces natural dynamic motion. A more detailed description and discussion of our symmetry-aware encoding (including the reflection axis, 3D camera augmentation, and the embedding design) is provided in our response to Reviewer R1qYS–Q2, which is significantly different from how prior works use symmetry priors (mostly by directly constraining the base shape).
>
>
> ---
>
> **W2**: *"The improvement from introduced components (e.g., Tab 3) are not that significant."*
>
> **A**:To more carefully assess the contribution of each component, we re-ran the ablation study under a unified hyperparameter setting on both our original datasets and the new Artemis dataset.
>
> 1. Quantitative Results:
>
>     Table 3: Ablation on the Online dataset:
>
>     | Method | **PSNR**$\uparrow$ | **SSIM**$\uparrow$ | **LPIPS**$\downarrow$ | **KID**$\downarrow$ | **FVD-F**$\downarrow$ | **FVD-D**$\downarrow$ |
>     | :--- | :---: | :---: | :---: | :---: | :---: | :---: |
>     | Random Init | 25.253 | 0.909 | 0.102 | 0.199 | 1071.278 | *1167.620* |
>     | w/o Deform | 24.522 | 0.898 | 0.095 | 0.173 | 1019.340 | 1355.544 |
>     | w/o Joint Anchors | 25.872 | *0.916* | 0.086 | 0.170 | 1025.340 | 1255.729 |
>     | w/o Symmetry Encoding | 25.666 | 0.913 | 0.094 | *0.158* | *965.699* | 1176.181 |
>     | w/o $L_{smooth}$ | *25.942* | **0.918** | **0.085** | **0.153** | 994.022 | 1203.356 |
>     | **Full (Ours)**| **25.961** | **0.918** | **0.085** | 0.159 | **959.232** | **1148.435** |
>
>     Our full model achieves the best or tied-best performance on most metrics, and clearly improves over the random initialization and “w/o Deform” variants across all metrics.
>
>     Table 4: Ablation on Artemis, where ground-truth views are available. We simulated a monocular video setting by training on a sequence of single views and evaluating on held-out views (resulting in a 1:35 / 1:71 train/test split). we also report D-3DGS for reference:
>
>     | Method | PSNR $\uparrow$ | SSIM $\uparrow$ | LPIPS $\downarrow$ |
>     | :--- | :---: | :---: | :---: |
>     | D-3DGS | 20.050 | 0.903 | 0.103 |
>     | Random Init | 22.189 | 0.889 | 0.113 |
>     | w/o Deform | 23.037 | 0.931 | **0.071** |
>     | w/o Joint Anchors | 23.059 | 0.896 | 0.104 |
>     | w/o Symmetry Encoding | 21.515 | 0.920 | 0.090 |
>     | w/o $L_{smooth}$ | 22.215 | 0.859 | 0.130 |
>     | **Ours (Full)** | **24.113** | **0.937** | **0.071** |
>
>     On Artemis, our full model outperforms all variants and D-3DGS in PSNR/SSIM, while matching the best LPIPS.
>
> 2. Analysis: The results demonstrate that each component contributes significantly to either reconstruction quality or system stability:
>
>     - Deformation Module: On the Online Dataset (Table 3), removing the deformation module (w/o Deform) causes a drop of -1.44 dB PSNR, proving it is critical for capturing nonlinear dynamics in real animals. On Artemis (Table 4), the impact is smaller due to the rigid nature of the CGI assets, but our Full Model still adapts best.
>
>     - Symmetry Encoding: Removing the symmetry encoding leads to a drastic drop on Artemis (-2.598 dB), suggesting that the proposed symmetry-aware design is crucial for stabilizing the reconstructed geometry.
>
>     - Robustness: We observed that ablated variants (Random Init, w/o Joint Anchors, w/o Smooth) failed to converge on 1–2 challenging videos, leading to severe artifacts and the degraded metrics shown above. This confirms that a key benefit of our proposed components includes improved stability and robustness.
>
> ---

---

> ### Author Response · Authors · 2025-11-25
> **Response to Reviewer GMuu (2)**
>
> **W3**: *"The visual results are not great. The reconstruction suffer from blurriness and appearance artifacts."*
>
> **A**: We appreciate the reviewer’s comment. In our experiments, we found that most unstable reconstructions and artifacts are caused by inaccuracies in the monocular camera estimates from the preprocessing stage. Our pipeline filters out obviously unstable cameras and augments the data and use symmetry mechanism to mitigate camera noise. Improving camera stability is therefore an important direction for our future work. In addition, our model explicitly recover geometrically consistent and natural motion, which can trade off some high-frequency sharpness for reduced flickering and implausible deformations.
>
>  Nevertheless, despite occasional blur in challenging sequences, our method achieves state-of-the-art results on multiple dataset, and yields more natural and stable 4D reconstructions in our qualitative comparisons (see our supplementary video).
>
>
> ---
>
> **W4**: *"Missing baseline and related work: it seems DreamMesh4d has more appealing result and should be compared against."*
>
>
> **A**: We thank the reviewer for bringing DreamMesh4D to our attention. We have added it as a baseline and updated the quantitative results.
>
> 1. Quantitative Comparison: The results (summarized below) reveal an interesting trade-off based on the nature of the video data:
>
>
>     | Datasets | | **DAVIS** | | | **Online** | | | **APTv2** | |
>     | :--- | :---: | :---: | :---: | :---: | :---: | :---: | :---: | :---: | :---: |
>     | **Input-View** | **PSNR**$\uparrow$ | **SSIM**$\uparrow$ | **LPIPS**$\downarrow$ | **PSNR**$\uparrow$ | **SSIM**$\uparrow$ | **LPIPS**$\downarrow$ | **PSNR**$\uparrow$ | **SSIM**$\uparrow$ | **LPIPS**$\downarrow$ |
>     | DreamMesh4D | 23.150 | 0.881 | 0.116 | 23.859 | 0.889 | 0.131 | **26.000** | **0.899** | **0.102** |
>     | **Ours** | **26.254** | **0.917** | **0.085** | **25.961** | **0.918** | **0.085** | *24.678* | *0.873* | *0.133* |
>     | **Novel-View** | **KID**$\downarrow$ | **FVD-F**$\downarrow$ | **FVD-D**$\downarrow$ | **KID**$\downarrow$ | **FVD-F**$\downarrow$ | **FVD-D**$\downarrow$ | **KID**$\downarrow$ | **FVD-F**$\downarrow$ | **FVD-D**$\downarrow$ |
>     | DreamMesh4D | 0.148 | 1154.038 | 2257.034 | *0.185* | 1518.163 | 2911.299 | *0.188* | **629.482** | **720.280** |
>     | **Ours** | **0.138** | **912.685** | **804.277** | **0.159** | **959.232** | **1148.435** | **0.170** | *991.472* | *930.146* |
>
>
> 2. Analysis of Performance:
>
>     DAVIS & Online: Our method consistently outperforms DreamMesh4D across all input-view and novel-view metrics. These datasets contain longer sequences with smooth animal motion and more moving cameras. In this regime, our method is effective at maintaining temporal consistency and accurate geometry, while DreamMesh4D exhibits unnatural motion artifacts (also visible in the supplementary video). This is consistent with the limitations discussed in the DreamMesh4D paper itself, where handling moving viewpoints remains challenging.
>
>     APTv2: DreamMesh4D performs strongly on APTv2. We attribute this to the dataset characteristics: sequences are very short (15 frames) with large inter-frame motion, and camera motion is limited. In such cases, the powerful generative prior of DreamMesh4D is advantageous for hallucinating missing views. Nevertheless, our approach still achieves the second-best results on APTv2 among all methods evaluated.
>
>     Efficiency: Our method takes about 16 minutes per sequence versus 50 minutes for DreamMesh4D on the same NVIDIA A6000 workstation and videos.

---

> > ### Author Response · Authors · 2025-11-25
> > **Response to Reviewer GMuu (3)**
> >
> > **Q1**: *It would help if the authors could highlight the major difference compared to existing works in writing, such as LASR and DreamMesh4d."*
> >
> >
> > **A**: We thank the reviewer for the suggestion. Here is a summary of the major distinctions:
> >
> >
> > - LASR [1] is a pioneering template-free method that reconstructs dynamic 3D shapes as explicit meshes, initialized from a sphere and optimized via video-consistency cues (optical flow, silhouettes) with a ResNet-18 backbone. However, it often struggles with complex topologies and high-frequency details.
> >    * In contrast, we use a 3D Gaussian Splatting representation initialized from Fauna, with a two-stage model for linear pose and nonlinear deformation, plus joint anchors and symmetry-aware encoding. This design avoids mesh topology constraints and allows us to capture more accurate geometry and volumetric details while mitigating camera noise.
> >
> >
> >  - DreamMesh4D [2] tackles a different problem setting: it is a 4D generation pipeline driven by a large image-to-3D prior (Zero123) and a hybrid SuGaR mesh–3DGS representation. Its appearance quality is therefore strongly influenced by the generative prior; as is common for such models, the generated textures can deviate from the true animal appearance (e.g., stylized or toy-like), which limits fidelity when reconstructing real videos (see our supplementary video).
> >
> >     * In contrast, we focus on 4D reconstruction of a given **real animal** from a single monocular video without relying on large generative priors. Moreover, DreamMesh4D’s hybrid mesh–3DGS representation is constrained by mesh topology. This often leads to relatively stiff motion. Our method instead uses a pure 3DGS representation with a two-stage deformation model, which allows more flexible deformation and avoids mesh topology constraints. As also discussed in their limitation section, their current pipeline assumes fixed cameras constrained by their generative prior. Our pipeline can handle more complex video scenes, and is also substantially more efficient in practice (≈16 min vs. ≈50 min per sequence)
> >
> >
> > [1] Yang, Gengshan, et al. "Lasr: Learning articulated shape reconstruction from a monocular video." Proceedings of the IEEE/CVF Conference on Computer Vision and Pattern Recognition. 2021.
> > [5] Li, Zhiqi, Yiming Chen, and Peidong Liu. "Dreammesh4d: Video-to-4d generation with sparse-controlled gaussian-mesh hybrid representation." Advances in Neural Information Processing Systems 37 (2024): 21377-21400.
> >
> >
> >
> > ---
> >
> > **Q2**: *"Compared to symmetry prior, image-to-3d models provides a stronger 3d prior and could help disentangle shape and motion."*
> >
> >
> > **A**: We agree that image-to-3D models can provide a strong 3D prior. In fact, our pipeline already leverages such a prior: we use Fauna (an image-to-3D quadrupeds model) to obtain an initial canonical mesh and articulation, and our pipeline is used on top of this initialization.
> >
> > Regarding large generative image-to-3D models (e.g., Trellis/Zero123-style priors), our experiments suggest that simply relying on a stronger generative prior is insufficient for our task (Specifically, GVFDiffusion is Trellis-based and DreamMesh4D is Zero123-based). Generative models tend to produce oversmoothed textures, plastic-like appearance, and stiff motion compared to our optimization-based method (see qualitative comparisons in the supplementary video).
> >
> > Importantly, our symmetry-aware encoding is orthogonal to the choice of 3D prior and could in principle also be integrated into generative image-to-3D pipelines. We view this as an interesting direction for future work.
> >
> > ---
> >
> > **Q3**: *"For evaluation, I found the current NVS metrics not convincing as those are distribution-level. Multiview dataset such as "Artemis: Articulated Neural Pets with Appearance and Motion Synthesis" can be an alternative to measure novel view view synthesis scores."*
> >
> >
> > **A**: We thank the reviewer for recommending the Artemis dataset. We agree that it serves as a more effective benchmark for assessing geometric accuracy than distribution-level metrics.
> >
> > As detailed in our response to W2 (please see Table 4 above), we have strictly followed your suggestion to evaluate our method on Artemis using standard, pixel-aligned novel view synthesis metrics (PSNR, SSIM, LPIPS) against Ground Truth. Qualitative comparison results are added to our supplementary video.
> >
> > ---
> >
> > We hope that our responses have satisfactorily addressed your concerns, and we would gladly take any additional questions you may have.

---

### Official Review · Reviewer_1qYS · 2025-11-01

**Soundness:** 3
**Presentation:** 3
**Contribution:** 2
**Rating:** 6
**Confidence:** 4

**Summary:**

This paper introduces a test-time optimization framework for generating time-variant 3D Gaussian Splatting (3DGS) representations of animals from a single monocular video. The method first leverages Fauna, an off-the-shelf model pretrained on large-scale animal data, to acquire a coarse 3D mesh reconstruction. This mesh then initializes a 3DGS field, which is subsequently refined through a joint-aware and symmetry-aware optimization process. The proposed method demonstrates state-of-the-art (SOTA) performance on established benchmarks.

**Strengths:**

1. The paper's methodology is logically structured, effectively leveraging animal-specific geometric priors to achieve high-quality reconstruction.
2. The proposed method proves effective, with experiments demonstrating strong performance in both seen-view and novel-view settings.
3. The manuscript is clearly written and easy to understand, supported by high-quality figures.

**Weaknesses:**

1. The method's reliance on a strong animal segmentation prior to remove background influence needs clarification. This dependency seems at odds with Figure 1, which depicts a complex scene with multiple similar animals (e.g., camels). The paper does not appear to detail how the method would distinguish or handle such challenging multi-entity scenarios, making the figure's inclusion potentially misleading.
2. The methodology requires further clarification on several key points:
•	The baseline Fauna model provides per-frame deformed meshes, which this method reportedly discards. Please clarify the rationale for omitting this information and explain its relationship (or lack thereof) to the modules proposed in this paper.
•	Line 198: The initialization of attributes for the 3D Gaussian representation is unclear. Please specify the method and any statistical distributions used (e.g., for random initialization).
•	Symmetry-Aware Encoding: Please clarify if this is implemented simply as 2D data augmentation. Furthermore, the axis of reflection (e.g., horizontal, vertical) should be explicitly stated.
•	Line 215: The notations $J$ and $K$ are used without prior definition.
•	Line 256: The notation $t$ appears to be overloaded. In Line 187, it seems to represent time, whereas here it likely refers to the optimization iteration. Please clarify and consider using distinct notation to avoid ambiguity.
3.  The experimental analysis is missing a comparison of computational and time costs against previous methods. Given that this is a test-time optimization approach, quantifying the efficiency-performance trade-off is essential for a complete evaluation of the method's practical utility.

**Questions:**

Please see the weaknesses.

---

> ### Author Response · Authors · 2025-11-25
> **Response to Reviewer 1qYS**
>
> We sincerely appreciate you taking the time to provide detailed and positive feedback on our paper. Here are our answers to your questions:
>
> ---
>
> **W1**: "The method's reliance on a strong animal segmentation prior to remove background influence needs clarification. This dependency seems at odds with Figure 1, which depicts a complex scene with multiple similar animals (e.g., camels). The paper does not appear to detail how the method would distinguish or handle such challenging multi-entity scenarios, making the figure's inclusion potentially misleading."
>
>
> **A:** We thank the reviewer for the careful reading of our paper. We clarify that our method is designed for **single-animal reconstruction**.
> Our data pipeline (Lines 371–377) preprocesses each video to isolate one target animal instance from the background and other animals. For clarity of our task visualization, we omitted the segmentation masks in Figure 1.
>
> Concretely, we distinguish and segment the target instance as follows:
>
> - **DAVIS:** We use the provided instance masks and process each annotated instance independently. The camel example in Figure 1 comes from DAVIS; the dataset only provides a mask for the foreground camel we reconstruct, while the other camels remain part of the background.
>
>
> - **Online videos:** We apply Grounded-SAM with the animal category name as the text prompt and keep the bounding box with the highest confidence as the input to SAM.
>
>
> - **APTv2:** We use the dataset’s ground-truth tracking annotations as bounding-box prompts for SAM to maintain consistent identity across frames.
> ---
>
>
>
> **W2**:
> **A**: We sincerely thank the reviewer for careful reading our paper. These questions help us to reduce the ambiguous parts in our original version. Below we address each point in turn.
>
>
> 1. *"The baseline Fauna model provides per-frame deformed meshes, which this method reportedly discards. Please clarify the rationale for omitting this information and explain its relationship (or lack thereof) to the modules proposed in this paper."*
>
>
>     - Our pipeline adopts a canonical deformation formulation, so we only need to initialize 3D Gaussians once in the canonical space. Using one Fauna-estimated rest-pose mesh is sufficient and avoids the inconsistencies of Fauna’s per-frame outputs (different vertex count, skinning weights, and potential failed poses). Moreover, Fauna’s deformed meshes are sometimes inaccurate (e.g., Fig. 3, fourth column), and our method learns its own deformation field, making per-frame Fauna meshes unnecessary.
>
>
> 2. *"Line 198: The initialization of attributes for the 3D Gaussian representation is unclear. Please specify the method and any statistical distributions used (e.g., for random initialization)."*
>
>
>     - We apologize for the lack of detail in the original text. For centers, we directly use mesh vertices. For orientations, we construct an orthonormal frame using the vertex normal and a randomly sampled tangent direction (standard normal distribution). For SH coefficients, the DC term is initialized with random RGB values (uniform distribution), and all higher-order SH terms are set to zero. We initialize scale based on vertex-connected edge lengths and use a fixed initialization of 0.9 for opacity. We acknowledge that the previous description was oversimplified and will update the main text and provide these details in the appendix.

---

> ### Author Response · Authors · 2025-11-25
>
> (Continued)
>
> 3. *"• Symmetry-Aware Encoding: Please clarify if this is implemented simply as 2D data augmentation. Furthermore, the axis of reflection (e.g., horizontal, vertical) should be explicitly stated."*
>
>
>     We thank the reviewer for pointing this out. We realized that our original description could be misleading and will revise the corresponding parts in Sec 3.2 and Sec 3.4.
>
>     - Axis of Reflection: The axis of reflection corresponds to the bilateral (sagittal) plane of the animal. In our coordinate system, this is explicitly the YoZ plane (i.e., flipping along the X-axis). We use horizontal flipping for the 2D data augmentation.
>
>
>     - Clarification on Implementation (Not just Data Augmentation): We clarify that our Symmetry-Aware Encoding is not simply 2D data augmentation. It is a specialized embedding strategy designed to mitigate 3D geometric drift.
>
>
>         * Contrast with Simple 2D Augmentation: We clarify that our Symmetry-Aware Encoding is not simply 2D data augmentation. It is a specialized strategy that involves explicit 3D camera symmetrization and a drift-aware embedding.
>             Our approach differs from standard augmentation in two key aspects:
>
>
>             * 3D Camera Augmentation: Unlike standard 2D augmentation, which operates solely on images, we explicitly augment the 3D camera positions by computing the symmetric extrinsic parameters ($C_{sym}$) relative to the reflection plane. Note: For the baselines, we did not include this 3D camera augmentation, strictly using standard 2D image augmentation only for a fair comparison.
>
>
>             * Encoding Strategy: If we simply used a basic encoding like $emb(t \cdot v)$ (where $v \in \{-1, 1\}$ indicates 2D image flip state and $t \in (0,1]$ to avoid aliasing at $t=0$), we would force the mismatched cameras to align. Note: We utilized this basic encoding setting in our ablation study ("w/o Symmetry Encoding") and applied it to our baselines.
>
>
>         * Our Motivation: We observed that the mathematically symmetric camera ($C_{sym}$) often deviates from the estimated camera of the flipped view ($C_{flip}$) due to estimation noise in the preprocessing step (Fauna). Directly forcing them to render the same view (corresponding to $I_{flip}$) causes 3D aliasing.
>
>
>        * Our Solution: We organize the augmented samples into two groups: the Original Group $\mathcal{V} _ {\text{orig}} = \{(I^t, C^t, P^t), (I^t _ {\text{flip}}, C^t _ {\text{sym}}, P^t _ {\text{flip}})\}$ and the Flipped Group $\mathcal{V}_{\text{flip}}= \{(I^t _ {\text{flip}}, C^t _ {\text{flip}}, P^t _ {\text{flip}}), (I^t, C^t _ {\text{flip,sym}}, P^t)\}$. We distinguish these groups using an indicator $m \in \{-1, 1\}$, which represents the source of the 3D camera estimation. We designed the symmetry-aware encoding $emb(t \cdot m) \oplus v$, which maps the input time $t$ into four distinct subspaces. This mechanism allows the network to learn the shared canonical geometry while disentangling and absorbing the camera-dependent noise.
>
>
> 4. *"• Line 215: The notations $J$ and $K$ are used without prior definition."*
>
>     We apologize for the omission. In this context, $J$ denotes the number of joints, and $K$ denotes the dimensionality of the keys and values in the subsequent cross-attention block. We will update the revision to introduce these notations.
>
>
> 5. *"• Line 256: The notation $t$ appears to be overloaded. In Line 187, it seems to represent time, whereas here it likely refers to the optimization iteration. Please clarify and consider using distinct notation to avoid ambiguity."*
>
>     Thank you for pointing out this ambiguity. We will update our paper to consistently use $t$ to represent time and $i$ to denote the optimization iteration.
>
> ---
>
> **W3**: *"The experimental analysis is missing a comparison of computational and time costs against previous methods. Given that this is a test-time optimization approach, quantifying the efficiency-performance trade-off is essential for a complete evaluation of the method's practical utility."*
>
>
> **A**: Thank you for pointing this out. We have reported the time cost comparison on the same NVIDIA A6000 workstation and videos:
>
> | Method | D-3DGS | GVFDiffusion | GART | GART (w/ TTO) | DreamMesh4D | **Ours** |
> | :--- | :---: | :---: | :---: | :---: | :---: | :---: |
> | Time (min) | 3.34 | 5.26 | 8.76 | 21.86 | 50.38 | 16.84 |
>
> Our method (~16.8 mins) achieves a balance between computational cost and reconstruction fidelity.
> Compared with the fastest methods (e.g., D-3DGS and GVFDiffusion), our approach provides significantly better reconstruction quality. Compared to the concurrent work DreamMesh4D (50.38 mins), our method is significantly more efficient. Crucially, this speed gain does not come at the cost of quality, as evidenced by the updated Tables 1 and 2.
>
>
> ---
>
> We sincerely hope that our answers have adequately addressed your concerns, and we are more than willing to answer any further questions.

---

### Official Review · Reviewer_PeBy · 2025-11-01

**Soundness:** 3
**Presentation:** 3
**Contribution:** 2
**Rating:** 4
**Confidence:** 3

**Summary:**

This paper introduces AnimalGS, an optimization-based framework for 4D animal reconstruction from monocular video using 3DGS as the explicit representation. Unlike previous methods based on category-specific templates or generative priors, the animal shape prior predicted by Fauna is used only as initialization, rather than as constraints.

The innovation of this paper mainly focuses on model design: 1. Joint-Aware achors for robut pose refinement; 2. Symmetry-aware temporal encoding for bilateral cues; Pose-guided deformation based on cross-attention between joint and gaussian features.

Experimental comparisons with Fauna, GART, D-3DGS, and GVF-Diffusion are provided to demonstrate the effectiveness of the proposed pipeline.

**Strengths:**

1. This paper proposes a template-free pipeline for 4D animal reconstruction with requiring dataset-level supervision.
2. The joint-anchor and symmetry encoding are interpretable designs, which stablizes optimization process.
3. The pose-conditioned deformation provides a clever constrain on 3DGS pipeline.
4. The paper is well-written and easy to understand.

**Weaknesses:**

1. Optimization-based methods require per-sample training, which is time-consuming.
2. Quantitative metrics are proxy-based without 3D ground truth, which cannot assess the geometric accuracy.
3. The Deformable-3DGS is not fairly compared in an identical setting: Initialize the Gaussians with Fauna and use RGB and Silhouette together for training. This critical result is missing.
4. The method still depends on Fauna for initialization; if Fauna fails catastrophically, it’s unclear how robust AnimalGS remains.

**Questions:**

Please address my concerns in the weakness part, and I will increase the score accordingly.

---

> ### Author Response · Authors · 2025-11-25
> **Response to Reviewer PeBy**
>
> **Reviewer PeBy:**
>
> We really appreciate all your constructive feedback and insightful questions for our papers. Here are our answers to your questions:
>
> ---
> **W1**: *"Optimization-based methods require per-sample training, which is time-consuming."*
>
> **A1**:
>   We appreciate your discussion regarding efficiency. We acknowledge that per-sample optimization is computationally intensive; however, this process is essential in our framework to ensure high-fidelity geometry and motion from monocular inputs while feed-forward methods often struggle to preserve.
>
>   Our full optimization takes about 16 minutes per sequence (measured on a single NVIDIA A6000). We believe this cost is reasonable given the observed improvements in geometric consistency, texture detail, and natural motion over other baselines, but we also view further reducing the optimization time as an important direction for future work.
>
>
> ---
>
> **W2**: *"Quantitative metrics are proxy-based without 3D ground truth, which cannot assess the geometric accuracy."*
>
>
> **A2**:  We appreciate this concern. We acknowledge that 2D proxy metrics on monocular videos have limitations. To rigorously assess geometric accuracy without 3D ground truth scans on real animals, we followed the advice of Reviewer GMuu and evaluated our method on the Artemis dataset [2].
>
>
> 1. Quantitative Geometric Validation (Artemis): Artemis provides high-fidelity CGI animals with 36/72 synchronized cameras. We trained our model on a monocular sequence and evaluated it on held-out, unseen viewpoints (training:test = 1:35 for most sequences, and 1:71 split for the *wolf* sequence). High performance on these unseen views is a direct indicator of accurate underlying geometry.
> As shown in the table below, our method significantly outperforms the baseline (reimplemented D-3DGS). This substantial gap quantitatively validates that our method recovers significantly more accurate geometry and texture than the baseline.
>
>
>     | Method | PSNR $\uparrow$ | SSIM $\uparrow$ | LPIPS $\downarrow$ |
>     | :--- | :---: | :---: | :---: |
>     | D-3DGS | 20.050 | 0.903 | 0.103 |
>     | **Ours (Full)** | **24.113** | **0.937** | **0.071** |
>
>
> 2. Perceptual Evaluation: Complementing the quantitative results, our User Study (as detailed in Sec. 4.3 and Supplementary A.3) across 4 dimensions (temporal stability, motion naturalness, appearance fidelity, shape integrity) further confirms that human evaluators consistently rate our geometric reconstruction as superior.

---

> > ### Author Response · Authors · 2025-11-25
> >
> > **W3**: *"The Deformable-3DGS is not fairly compared in an identical setting: Initialize the Gaussians with Fauna and use RGB and Silhouette together for training. This critical result is missing."*
> >
> > **A3**: We appreciate the reviewer’s comment on ensuring a fair comparison.
> >
> > 1. **Clarification on initialization.** We would like to clarify that in the original submission our D-3DGS baseline already used the same Fauna-based initialization as our method. We apologize for not stating this explicitly.
> >
> > 2. **Adding silhouette supervision to D-3DGS.** Our original baseline indeed lacked silhouette supervision. This was because the original D-3DGS rasterizer does not support alpha map rendering. To fully match our setting and address your concern, we:
> >
> >     * replace the original rasterizer with diff-gaussian-rasterization [3], which supports alpha rendering; and
> >
> >
> >     * retrain D-3DGS with joint RGB + silhouette supervision under exactly the same Fauna initializations, camera estimates, and loss weights as our method.
> >
> >
> > 3. Results: The updated comparison is shown below. With the added silhouette supervision, D-3DGS indeed improves over our previous baseline. However, our method still consistently outperforms this stronger baseline across all three datasets, both on input-view and novel-view reconstruction. Note: We have re-run our model using the same hyperparameters across all three datasets and the new Artemis dataset.
> >
> >     |  | | **DAVIS** | | | **Online** | | | **APTv2** | |
> >     | :--- | :---: | :---: | :---: | :---: | :---: | :---: | :---: | :---: | :---: |
> >     | **Input-view** | **PSNR**$\uparrow$ | **SSIM**$\uparrow$ | **LPIPS**$\downarrow$ | **PSNR**$\uparrow$ | **SSIM**$\uparrow$ | **LPIPS**$\downarrow$ | **PSNR**$\uparrow$ | **SSIM**$\uparrow$ | **LPIPS**$\downarrow$ |
> >     | **D-3DGS** | 25.776 | 0.905 | 0.100 | 25.659 | 0.912 | 0.096 | 23.636 | 0.850 | 0.144 |
> >     | **Ours**| 26.254 | 0.917 | 0.085 | 25.961 | 0.918 | 0.085 | 24.678 | 0.873 | 0.133 |
> >     | **Novel-view** | **KID**$\downarrow$ | **FVD-F**$\downarrow$ | **FVD-D**$\downarrow$ | **KID**$\downarrow$ | **FVD-F**$\downarrow$ | **FVD-D**$\downarrow$ | **KID**$\downarrow$ | **FVD-F**$\downarrow$ | **FVD-D**$\downarrow$ |
> >     | **D-3DGS** | 0.199 | 1192.015 | 980.747 | 0.231 | 1244.444 | 1268.110 | 0.262 | 1181.275 | 1112.902 |
> >     | **Ours**| 0.138 | 912.685 | 804.277 | 0.159 | 959.232 | 1148.435 | 0.170 | 991.472 | 930.146 |
> >
> >
> >     Together with the Artemis results in A2, these experiments show that under a strictly fair setting, our method still achieves better 4D reconstruction quality than D-3DGS.
> >
> >
> >     [3] https://github.com/slothfulxtx/diff-gaussian-rasterization
> >
> > ---
> >
> >
> > **W4**: *"The method still depends on Fauna for initialization; if Fauna fails catastrophically, it’s unclear how robust AnimalGS remains."*
> >
> >
> > **A4**: We thank the reviewer for raising this robustness discussion. We have analyzed the robustness of our method against Fauna failures from two perspectives:
> >
> > 1. **Robustness against Shape Failure**: Even when Fauna initializes with a misaligned or inaccurate mesh, our optimization pipeline effectively corrects these errors.
> >
> >     * *Qualitative evidence*: As shown in Fig. 3 (Column 4), despite the initial Fauna mesh being significantly misaligned with the target animal, our method refines both shape and motion to match the input video.
> >
> >     * *Quantitative evidence*: To simulate an extreme “total geometric failure”, we have removed the Fauna prior and start from random Gaussians position in our ablation study (“Random Init”). Even under this setting, our method still converges to reasonable 4D reconstructions and clearly improves over this baseline (D-3DGS) on Artemis:
> >
> >         | Method | PSNR $\uparrow$ | SSIM $\uparrow$ | LPIPS $\downarrow$ |
> >         | :--- | :---: | :---: | :---: |
> >         | Random Init | 22.189 | 0.889 | 0.113 |
> >         | **Ours (Full)** | **24.113** | **0.937** | **0.071** |
> >
> >
> > 2. **Robustness against Camera Failure**: Empirically, we also observe that the final quality is more sensitive to camera estimation than to the exact Fauna shape. This is consistent with most monocular 4D reconstruction pipelines: while moderate camera jitters can be refined, recovering from catastrophic trajectory failures remains an open challenge.
> > In our current pipeline, we mitigate such effects by clustering and filtering unstable camera estimates during preprocessing, and by using symmetry-aware encoding combined with bilateral augmentation to partially compensate for removed frames. While more sophisticated joint optimization of cameras and geometry (e.g., an additional camera-refinement module) could further improve robustness, it would also increase test-time cost. We therefore view designing camera-aware yet efficient optimization schemes as an orthogonal and important direction for future work.
> > ---
> >
> > We hope that our replies have clarified your concerns, and we remain happy to respond to any additional questions.

---

### Official Review · Reviewer_YxkL · 2025-11-01

**Soundness:** 2
**Presentation:** 2
**Contribution:** 2
**Rating:** 4
**Confidence:** 5

**Summary:**

The paper proposes AnimalGS, a test-time optimization (TTO) framework that reconstructs 4D animal geometry and motion from a single monocular video using a canonical 3D Gaussian Splatting (3DGS) representation. The pipeline (i) initializes from a coarse FAUNA prior (single-image category-agnostic animal model), (ii) performs pose refinement via learnable joint anchors with a symmetry-aware temporal encoding, and (iii) applies pose-guided non-rigid deformation to capture residual motion/appearance. The method optimizes photometric/silhouette losses plus a normal-smoothness regularizer. Experiments on DAVIS/Online/APTv2 report improved input-view PSNR/LPIPS and novel-view KID/FVD vs. Fauna, GART, D-3DGS, and a diffusion baseline “GVF-Diffusion (Trellis-based).” The authors claim strong perceptual quality and present a small user study.

**Strengths:**

Addresses a high-impact problem: single-video 4D animal reconstruction.

The two-stage TTO with symmetry cueing is practical and fits 3DGS well; ablations indicate each piece helps.

Shows consistent improvements over baseline TTO/3DGS choices reported in the paper.

Implementation appears competent; real-time 3DGS rendering is attractive for downstream use.

**Weaknesses:**

Visual quality not consistently convincing: Even in curated figures, limbs and fine appendages exhibit shape drift/temporal wobble, and small non-rigid details are often smoothed out; some sequences look over-regularized (likely from the normal-smoothness phase). The paper does not show challenging real-world clips with fast motion/occlusion and admits remaining failure modes (head/tail subtleties).


Missing strong baselines for direct 3D/4D generation: No side-by-side against Hunyuan3D (image→3D) or Trellis-based pipelines (structured 3D latents). The paper only evaluates GVF-Diffusion as a proxy; that is insufficient—you should compare reconstruction fidelity and geometric consistency vs. (i) Hunyuan3D (image→3D + pose retarget), (ii) Trellis+render-supervised 4D generation, and (iii) a hybrid pipeline (Hunyuan3D init → your TTO).

Under-analyzed comparisons to the articulated-reconstruction literature the community expects:

DreaMo (single casual video → articulated model and motion)

LIMR (Learning Implicit Representation for Reconstructing Articulated Objects)

S3O (dual-phase dynamic shape + skeleton from single video)

PAD3R (pose-aware dynamic reconstruction from casual videos)

MagicArticulate (prepare static models for articulation)
Even if these target broader articulated objects (not only animals), their architectural choices, priors, and metrics are highly relevant. A clear, protocol-matched comparison (or at least cross-dataset transfer) is needed to justify the claimed advantages.

Over-reliance on priors without robustness study: Replace or degrade FAUNA/camera priors; inject mask/trajectory noise; quantify sensitivity.

Assumptions not stress-tested: What if first frames are not static? Provide an automatic canonical-frame detection ablation.

Evaluation scope: Mostly short clips, restrained motions, and curated species. Add unconstrained YouTube sequences, long clips, severe occlusions, and small articulators.

**Questions:**

Hunyuan3D / Trellis: Can you provide direct, protocol-matched comparisons? For Hunyuan3D, try (a) image→3D per key frame + tracking; (b) image→3D initialization followed by your TTO; (c) evaluate fidelity vs. plausibility trade-offs. For Trellis, evaluate a structured latent 4D pipeline with your same datasets/metrics.

Articulated-reconstruction baselines: Please add DreaMo, LIMR, S3O, PAD3R, MagicArticulate to discussion.

Robustness to priors: How does performance change if FAUNA predictions are noisy/misaligned (pose jitter, wrong joints), masks are imperfect, or cameras are biased? Provide degradation curves.

Canonical-frame assumption: Show results when the first frames contain motion; can you auto-select canonical segments?

Runtime/memory: Report training steps, seconds/iteration, Gaussians count vs. resolution, and Ablate λsmooth schedule; compare to GART / D-3DGS under identical hardware.

Failure analysis: Provide videos where tails/ears/paws fail; diagnose whether pose refinement or deformation is the bottleneck.

Physical plausibility: Do you enforce joint limits or bone-length constraints? If not, quantify bone-length variance over time.

---

> ### Author Response · Authors · 2025-11-25
> **Response to Reviewer YxkL**
>
> Thank you for providing such detailed comments on our study.
>
> ---
>
> **Q1** *"Hunyuan3D / Trellis: Can you provide direct, protocol-matched comparisons? For Hunyuan3D, try (a) image→3D per key frame + tracking; (b) image→3D initialization followed by your TTO; (c) evaluate fidelity vs. plausibility trade-offs. For Trellis, evaluate a structured latent 4D pipeline with your same datasets/metrics."*
>
>
> We appreciate the reviewer’s suggestion to compare against strong generative baselines. To address this, we have incorporated two state-of-the-art methods: GVF-Diffusion (Trellis-based) and DreamMesh4D (Zero123-based).
> We provide side-by-side visual comparisons in the Supplementary Material, and summarize the quantitative performance below:
>
> | Datasets | | **DAVIS** | | | **Online** | | | **APTv2** | |
> | :--- | :---: | :---: | :---: | :---: | :---: | :---: | :---: | :---: | :---: |
> | **Input-View** | **PSNR**$\uparrow$ | **SSIM**$\uparrow$ | **LPIPS**$\downarrow$ | **PSNR**$\uparrow$ | **SSIM**$\uparrow$ | **LPIPS**$\downarrow$ | **PSNR**$\uparrow$ | **SSIM**$\uparrow$ | **LPIPS**$\downarrow$ |
> | GVFDiffusion|  16.419 | 0.836 | 0.174 | 16.835 | 0.857 | 0.167 | 14.820 | 0.778 | 0.256 |
> | DreamMesh4D | 23.150 | 0.881 | 0.116 | 23.859 | 0.889 | 0.131 | **26.000** | **0.899** | **0.102** |
> | **Ours** | **26.254** | **0.917** | **0.085** | **25.961** | **0.918** | **0.085** | *24.678* | *0.873* | *0.133* |
> | **Novel-View** | **KID**$\downarrow$ | **FVD-F**$\downarrow$ | **FVD-D**$\downarrow$ | **KID**$\downarrow$ | **FVD-F**$\downarrow$ | **FVD-D**$\downarrow$ | **KID**$\downarrow$ | **FVD-F**$\downarrow$ | **FVD-D**$\downarrow$ |
> | GVFDiffusion | *0.145* | 1872.192 | 1270.189 | *0.179* | 1673.419 | 1387.732 | 0.274 | 1715.471 | 1575.461 |
> | DreamMesh4D | 0.148 | 1154.038 | 2257.034 | *0.185* | 1518.163 | 2911.299 | *0.188* | **629.482** | **720.280** |
> | **Ours** | **0.138** | **912.685** | **804.277** | **0.159** | **959.232** | **1148.435** | **0.170** | *991.472* | *930.146* |
>
> Our method significantly outperforms both GVF-Diffusion and DreamMesh4D on the DAVIS and Online datasets. On APTv2, DreamMesh4D achieves the best input-view metrics, while our method is the second best, remaining competitive on novel-view metrics. We attribute DreamMesh4D’s strong performance on APTv2 to the dataset characteristics: very short sequences (≈15 frames) with large inter-frame motion and minimal camera movement. In such settings, generative priors are effective at hallucinating missing views. In contrast, on datasets containing longer sequences, smooth motion, and moving cameras, our reconstruction-oriented test-time optimization consistently achieves higher fidelity, temporal stability, and geometric accuracy than generative methods.
>
> ---
>
> **Q2** *"Articulated-reconstruction baselines: Please add DreaMo, LIMR, S3O, PAD3R, MagicArticulate to discussion."*
>
> **A**:
> Thank you for pointing out these additional articulated-reconstruction works. We have reviewed all of them and found that:
>
> DreaMo, LIMR, S3O, and PAD3R do not provide publicly available or runnable implementations. Although some of them have GitHub repositories, they do not contain the code required to reproduce the methods. This makes protocol-matched comparisons on our datasets infeasible.
>
> MagicArticulate targets a fundamentally different problem—rigging static 3D assets for articulation—not reconstructing 4D geometry and motion from single-view videos. Its assumptions and outputs are therefore not comparable to video-based 4D reconstruction methods.
>
> While these works are conceptually relevant in a broad sense, their settings and availability make direct comparison impractical. We will add a short discussion in the revised related-work section to clarify their relation to our method.
>
> ---
>
> **Q3** *"Robustness to priors: How does performance change if FAUNA predictions are noisy/misaligned (pose jitter, wrong joints), masks are imperfect, or cameras are biased? Provide degradation curves."*
>
> **A**: Thank you for bringing up these discussion topics. We agree that robustness to imperfect priors is important. Our current experiments already cover several scenarios where the priors are noisy or inaccurate:
>
> • *"Pose jitter"*.
> Fauna is a single-image model and therefore provides no temporal consistency; pose jitter is common. Our test-time optimization naturally smooths this noise, and we observe that small pose errors do not noticeably degrade the final result, as the deformation and refinement stages absorb such variations.

---

> ### Author Response · Authors · 2025-11-25
>
> (Continued)
>
> • *"Imperfect masks"*.
> For in-the-wild videos, masks are obtained using Grounded-SAM. Occasional mask inaccuracies are tolerated because our model is optimized temporally and uses multiple frames. However, when regions of the animal are persistently occluded (e.g., self-occlusion or external objects), silhouette-guided reconstruction cannot hallucinate missing geometry—this is an inherent limitation of reconstruction-based methods. We will mention this in the revised manuscript.
>
> • *"Camera bias / camera drift"*.
> Empirically, we find that reconstruction quality is more sensitive to camera estimation errors than to the Fauna shape. While our pipeline cannot fully recover from completely incorrect camera poses, it can handle moderate instability. We mitigate camera drift by (i) clustering and filtering obviously unstable Fauna cameras during preprocessing, and (ii) using symmetry-aware encoding with bilateral augmentation, which helps absorb camera noise among augmented images. This is also reflected in the improved stability seen in our ablations.
>
> ---
>
>
> **Q4** *"Canonical-frame assumption: Show results when the first frames contain motion; can you auto-select canonical segments?"*
>
> **A**: *"In our current pipeline, we directly use the Fauna prediction from the first frame as the initialization. Frames contain obviously incomplete animal will be filtered by comparing masks in data preprocessing step, making this choice reliable in practice."*
>
> We agree that automatically selecting the most canonical frame—or jointly estimating a canonical shape—would further improve robustness and provide a better initialization for 3DGS, especially for cases where the initial frames contain motion or partial occlusions. This is a valuable direction, and we will consider incorporating canonical-frame selection in future work.
>
> **Q5** : *"Runtime/memory: Report training steps, seconds/iteration, Gaussians count vs. resolution, and Ablate λsmooth schedule; compare to GART / D-3DGS under identical hardware."*
>
> We thank the reviewer for highlighting the importance of reporting detailed runtime and optimization statistics. Below, we summarize the requested information:
>
> * Runtime comparison (same hardware: NVIDIA A6000)
>
>     | Method | D-3DGS | GVFDiffusion | GART | GART (w/ TTO) | DreamMesh4D | **Ours** |
>     | :--- | :---: | :---: | :---: | :---: | :---: | :---: |
>     | Time (min) | 3.34 | 5.26 | 8.76 | 21.86 | 50.38 | 16.84 |
>
> * *"Iteration counts"* (also stated in Sec. 4.2 of our paper)
>
>   20k iterations for DAVIS / Online / Artemis videos; 10k iterations for APTv2.
>   D-3DGS is trained with 20k iterations on all datasets. All Gaussian parameters follow the learning rate schedule of 3DGS, and other modules are optimized with a single Adam using an exponentially decaying learning rate from $8\times10^{-4}$ to $1.6\times10^{-4}$.
>
>
> * *"Gaussians count vs. resolution"*
>
>   We report the average and median numbers of Gaussians across datasets (all experiments are conducted at resolution 512 x 512):
>
>   |                | DAVIS     | Online    | APTv2     | Artemis   |
>   |----------------------|-----------|-----------|-----------|-----------|
>   | Gaussians count              | 45606.6 | 53323.8 | 43297.3 | 25649.9 |
>   | Median               | 46155.0  | 37343.0  | 37003.0  | 24399.5  |
>
>   We attribute the lower Gaussian count on Artemis to two main factors:
>
>   - Accurate cameras: Artemis provides ground-truth camera poses. As a result, the optimization does not need to spawn additional Gaussians to compensate for reprojection inconsistencies caused by camera-estimation noise (DAVIS/Online/APTv2).
>
>   - Simpler motion and deformation: Real animals exhibit complex non-linear soft-tissue dynamics and challenging motion changes, requiring a denser 3DGS representation. In contrast, CGI assets in Artemis are animated via rigid LBS with limited non-rigid deformation, making them sufficiently representable with fewer Gaussians.
>   - Real animals exhibit complex soft-tissue deformation and high-frequency motion variations, which require a denser representation. In contrast, the CGI assets in Artemis follow rigid LBS animation with much simpler non-rigid behavior, allowing faithful reconstruction with fewer Gaussians. This aligns with our “w/o Deform” ablation, where deformation plays a smaller role on Artemis due to its simpler motion.
>
> * *"compare to GART / D-3DGS under identical hardware."*
>
>     All baseline methods were run on the same NVIDIA A6000 workstation to ensure a fair comparison.

---

> ### Author Response · Authors · 2025-11-25
>
> **Q6** *"Failure analysis: Provide videos where tails/ears/paws fail; diagnose whether pose refinement or deformation is the bottleneck."*
>
> **A** We thank the reviewer for the suggestion. We will provide failure cases in our supplementary. Our ablations already help diagnose which part of the pipeline is responsible for failures.
>
>  * Deformation module.
>     We evaluate the impact of removing the deformation branch (“w/o Deform”) on both real videos (Online) and the CGI multi-view dataset (Artemis):
>
>     Online dataset:
>     Removing deformation causes a clear performance drop (−1.44 dB PSNR), indicating that the deformation module is essential for modeling the nonlinear soft-tissue dynamics and fine local motions present in real animals:
>
>     | Method | **PSNR**$\uparrow$ | **SSIM**$\uparrow$ | **LPIPS**$\downarrow$ | **KID**$\downarrow$ | **FVD-F**$\downarrow$ | **FVD-D**$\downarrow$ |
>     | :--- | :---: | :---: | :---: | :---: | :---: | :---: |
>     | w/o Deform | 24.522 | 0.898 | 0.095 | 0.173 | 1019.340 | 1355.544 |
>     | **Full (Ours)**| **25.961** | **0.918** | **0.085** | 0.159 | **959.232** | **1148.435** |
>
>     Artemis dataset:
>     On Artemis, the degradation is much smaller:
>
>     | Method | PSNR $\uparrow$ | SSIM $\uparrow$ | LPIPS $\downarrow$ |
>     | :--- | :---: | :---: | :---: |
>     | w/o Deform | 23.037 | 0.931 | **0.071** |
>     | **Ours (Full)** | **24.113** | **0.937** | **0.071** |
>
>     This difference matches the nature of the datasets:
>     CGI characters follow rigid LBS animation with limited non-rigid deformation, so removing the deformation module has a smaller effect. Real animals exhibit rich non-linear body motion, where deformation is necessary.
>
> * Pose refinement module.
>     While we did not remove the pose module directly—because it is an upstream component—D-3DGS naturally serves as a proxy for “deformation-only” reconstruction (no pose refinement). Across all datasets, D-3DGS struggles to maintain shape integrity, especially around limbs and extremities, where motion is large and varies per joint.
>     This suggests that pose refinement provides the coarse articulated motion, reducing the burden on the deformation module and preventing drift or distortion in fine structures.
>
> ---
>
> **Q7** *"Physical plausibility: Do you enforce joint limits or bone-length constraints? If not, quantify bone-length variance over time."*
>
> **A** Thank you for raising this question. In our pipeline, we do not explicitly enforce joint-angle limits or bone-length constraints. This is because, after voxelizing the skinning weights during initialization, we adopt a skeleton-free LBS formulation following Liao et al. [1]. This design allows the joint transformations to be optimized in a more flexible and continuous manner rather than being restricted to a fixed articulated hierarchy.
>
> [1] Liao, Zhouyingcheng, et al. "Skeleton-free pose transfer for stylized 3d characters." European Conference on Computer Vision. Cham: Springer Nature Switzerland, 2022.

---

### Author Response · Authors · 2025-12-02
**To All Reviewers**

We would like to sincerely thank all reviewers for the time, effort, and thoughtful feedback provided on our submission. Your comments have greatly helped us improve the clarity, technical rigor, and overall quality of the paper.

Unfortunately, due to the unexpected interruption of the discussion period, we were unable to complete our discussion with you or receive your follow-up responses. We genuinely regret not having the opportunity to engage in further dialogue.

We hope that the additional experiments, analyses, and revisions we have included during the rebuttal period adequately address your concerns. Thank you again for your careful evaluation and constructive guidance.

---

### Author Response · Authors · 2025-12-02
**To the Area Chair**

We deeply appreciate the Area Chair’s extra time and effort in managing our submission, particularly under the unusual conditions that affected the rebuttal process. During the rebuttal, we have conducted extensive experiments to address all reviewers' concerns. To support your final decision-making and ease the workload, we have summarized the key discussions and updates below:

---

**1. Expanded Experiments**

- Re-implemented D-3DGS with Fauna initialization and silhouette supervision for a fair comparison; our method still outperforms this strengthened baseline.
- Added comparisons with the concurrent DreamMesh4D, showing better texture fidelity, more stable motion, and faster optimization.
- Added a full ablation on the Artemis (multi-view CGI) dataset, including D-3DGS results.
- Re-ran our method with unified hyperparameters across online videos, DAVIS, Artemis, and APTv2 (10k iters).
- Added new optimization-time comparisons across methods, and Gaussian-count analysis across datasets for our method.

---

**2. Supplementary Video Updates**

- Added comparisons with DreamMesh4D on real videos
- Added comparisons with re-implemented D-3DGS and DreamMesh4D on Artemis.
- Reorganized visualization layout for clarity and consistency.

    *We highly encourage viewing the updated supplementary video, which clearly demonstrates the task and the performance of our approach.*

---

**3. Paper Updates**

- Corrected ambiguous descriptions and notations (Sec. 3).
- Expanded data preprocessing details (Sec. 4.1).
- Updated quantitative tables (Tables 1–3) and added new tables (Tables 4–6: optimization time, Artemis comparison, Gaussian counts).
- Expanded visual comparisons (Figs. 3 & 5) and analysis (Sec. 4.3).
- Improved discussion of limitations (Sec. 4.4 and Fig. 6).
- Added detailed Gaussian initialization (Appendix A.2).
- Moved the user study to Appendix A.4 (still provided as supplementary evidence).
- Added Gaussian count analysis (Appendix A.5).

All changes in the paper are highlighted in magenta.

---

Thank you again for your time and consideration.

---

### Meta-Review · Area_Chair_wK8D · 2026-01-07

**Summary:**

The paper received somewhat mixed ratings (4, 4, 4, 6).

Reviewers recognized the challenging nature of the target problem, single-video 4D reconstruction, and noted that the paper demonstrates improved performance over compared approaches under the presented evaluation setup.

While reviewers provided several detailed comments, the AC identified three main concerns: (1) limited practical performance, particularly in the qualitative evaluations, where the results are not convincingly impressive or clearly superior to competing methods; (2) the complexity of the proposed approach, which relies heavily on multiple preprocessing components, such as FAUNA initialization and camera estimation; and (3) similarities to existing work on 4D reconstruction.

**Reviewer Concerns:**

During the authors’ response stage, the authors made substantial efforts to address the reviewers’ concerns.

Following the feedback, the authors provided additional experiments with important competing methods, evaluated the approach on a new dataset (e.g., artemis), performed further ablation studies, and provided additional qualitative video results.

In the quantitative evaluations, the AC found that the proposed method demonstrates advantages over existing work, achieving improved performance metrics at certain levels.

**Reviewer Scores:**

The AC carefully reviewed the paper, the reviewers’ original comments, the authors’ responses, and the response summary. Overall, the AC believes that the reviewers’ scores would likely remain unchanged, even after considering the authors’ thorough rebuttal.


The primary reason is the limited qualitative performance of the proposed method. In the qualitative evaluations, the results do not clearly stand out compared to competing approaches. For instance, in visual outputs, methods such as DreamMesh4D and GVFDiffusion appear visually more compelling, and it is difficult to conclude that the proposed method noticeably outperforms them. Although these competing methods achieve lower quantitative scores, largely due to deviations in color and texture fidelity relative to the original inputs, this reflects a trade-off. These methods exhibit fewer artifacts and less blurring by sacrificing texture similarity, which raises questions about the appropriate balance between perceptual quality and quantitative metrics.


While the AC appreciates the authors’ extensive quantitative evaluations, the reported performance differences are generally marginal and align with the impressions from the qualitative results. For example, in the table presented in the author's response, the PSNR scores on the “Online” DB  for D-3DGS versus the proposed method are 25.6 versus 25.9, and on “APTv2” they are 23.6 versus 24.6. Similarly, in the provided quant. tables, it remains difficult to conclude that the proposed method significantly outperforms the competitors.
More importantly, these quantitative metrics serve only as proxies for real-world performance. The qualitative results suggest that the proposed method does not represent a substantial breakthrough compared to existing approaches.


From a methodological perspective, the AC reached a similar conclusion. While the proposed approach introduces certain design choices and implementation details, it does not demonstrate a clear and significant advancement beyond existing test-time optimization techniques, which is consistent with the reviewers’ comments.

Based on these considerations, the AC concludes that the current version of the paper does not exceed the acceptance threshold.

The AC views the paper as borderline. It demonstrates methodological refinements and incremental numerical improvements. However, given the presence of several closely related approaches in this domain, stronger evidence is needed. The paper could reach an acceptable level if it were able to demonstrate more convincing and visually compelling results. This is particularly important in the current research era, where advances in video models provide alternative solutions for 4D reconstruction from a single image or video, by enabling high-quality novel view synthesis and motion control, raising the bar for perceptual and practical impact compared to previous years.

---

### Decision · Program_Chairs · 2026-01-26

Reject